# ARTICLES

## OPEN
# Molecular spikes: a gold standard for single-cell RNA counting

Christoph Ziegenhain [1,2], Gert-Jan Hendriks [1,2], Michael Hagemann-Jensen [1] and Rickard Sandberg [1✉]

Single-cell sequencing methods rely on molecule-counting strategies to account for amplification biases, yet no experimental strategy to evaluate counting performance exists. Here, we introduce molecular spikes—RNA spike-ins containing built-in unique molecular identifiers (UMIs) that we use to identify critical experimental and computational conditions for accurate RNA counting in single-cell RNA-sequencing (scRNA-seq). Using molecular spikes, we uncovered impaired RNA counting in methods that were not informative for cellular RNA abundances due to inflated UMI counts. We further leverage molecular spikes to improve estimates of total endogenous RNA amounts in cells, and introduce a strategy to correct experiments with impaired RNA counting. The molecular spikes and the accompanying R package UMIcountR (https://github.com/cziegenhain/UMIcountR) will improve the validation of new methods, better estimate and adjust for cellular mRNA amounts and enable more indepth characterization of RNA counting in scRNA-seq.

ScRNA-seq is used widely to dissect cellular states, types and trajectories[1]. Common to many single-cell technologies are counting strategies to mitigate the overcounting of amplicons derived from each RNA or DNA molecule. Typically, a random sequence, or UMI, is added via adapter oligos before DNA amplification and sequencing[2], and this strategy has become standard for RNA counting in single cells[3–6]. Despite the widespread use of UMIs, no experimental strategies exist that can be used to systematically quality control (QC) counting accuracy in new single-cell methods or variations in chemistries used. Furthermore, errors within the barcodes during amplification and sequencing necessitate subsequent ad hoc computational correction strategies. Several approaches for UMI error corrections to estimate RNA molecule counts have been proposed[7–9], but so far there are no experimental ground-truth datasets enabling standardized benchmarking. Here, we developed mRNA spike-ins that carry a high diversity random sequence (an internal UMI), which we use to assess the RNA counting accuracy of popular scRNA-seq methods and computational correction strategies. These spikes can be used to count RNAs correctly even in the absence of UMIs or when UMI-based counting had been inflated experimentally, and to estimate the mRNA amounts in cells.

## Results

**Establishing molecular spikes.** Randomized synthetic DNA sequences with minimal overlap to the human and mouse genomes were cloned into pUC19, together with a T7 promoter and a poly-A tail consisting of 30 adenine nucleotides (Fig. 1a). Oligonucleotide libraries carrying 18 random nucleotides were inserted either into the 5′ or 3′ region of the synthetic sequence to construct the spike-UMI (spUMI) of the 5′ and 3′ molecular spike, respectively (Fig. 1b). The resulting plasmid libraries were then used for in vitro transcription to produce molecular spike RNA pools (Fig. 1a). To test the produced spikes, we added the 5′ molecular spike to single HEK293FT cells and prepared Smart-seq3 libraries[6]. The spUMIs from the molecular spike sequences were extracted from aligned reads, and we similarly extracted the standard UMI sequence

introduced on the Smart-seq3 template-switching oligo. We verified that the 18 nucleotide (nt) spUMI was indeed predominantly random (Extended Data Fig. 1a). To counteract PCR and sequencing errors within the spUMIs on the molecular spikes, we investigated the appropriate error-correction strategy. To this end, for each molecular spike spUMI, we calculated the minimum edit distance (hamming distance) to the closest sequence within the cell and to 1,000 randomly sampled molecular spike spUMIs from other cells. This analysis demonstrated that the 18 nt spUMIs often showed an enrichment of spUMIs with one or two base errors within cells (Extended Data Fig. 1b). Moreover, random sampling of sequences of 18 nt in length is unlikely to yield collisions in sequence space (~68.7 billion sequences) at a hamming distance of 2 nt. Therefore, we used a hamming distance of 2 nt to infer the exact number of molecular spike spUMIs present in each cell for the remainder of the experiments in this study, and we further excluded spUMIs that were over-represented across cells (Methods) to remove potential biases (Extended Data Fig. 1c). We estimated the complexity of the total 5′ molecular spikes to 3.2 million by fitting an asymptotic nonlinear model to the number of observed spUMIs sequences across cells (Fig. 1c).

**Investigating counting performance in scRNA-seq methods.** Having validated the randomness and complexity of the spUMI, we investigated the RNA counting accuracy of single-cell methods, starting with Smart-seq3 (ref. [6]). Since the copy numbers of added 5′ molecular spikes were very high, we sampled spike molecules from the range of expression levels typically found in HEK293FT cells (Extended Data Fig. 2a–c). The observed error-corrected Smart-seq3 counts closely followed ($r^2 = 0.99$) the molecular spike ground-truth (error-corrected spUMIs) (Fig. 1d), demonstrating the accuracy in RNA counting in single cells with Smart-seq3.

Next, we exemplified how the molecular spikes can properly diagnose inaccuracies in scRNA-seq library protocols by investigating altered Smart-seq3 conditions in which a residual RNA-based template-switching oligo (TSO) is allowed to prime during PCR

[1]Department of Cell and Molecular Biology, Karolinska Institute, Stockholm, Sweden. [2]These authors contributed equally: Christoph Ziegenhain, Gert-Jan Hendriks. ✉e-mail: Rickard.Sandberg@ki.se

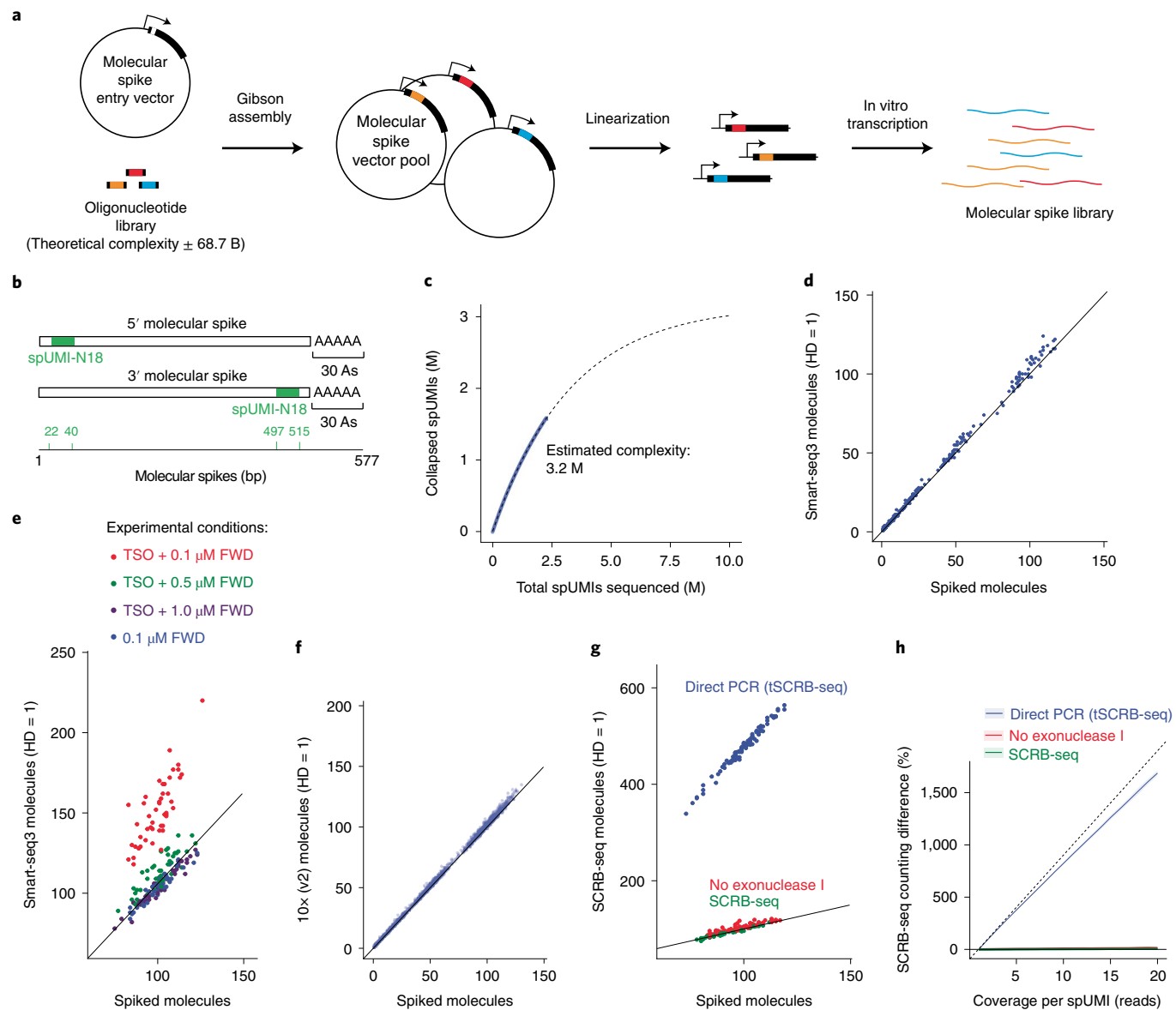

**Fig. 1 | Direct assessment of single-cell RNA counting using molecular spikes. a**, Schematic of cloning strategy of molecular spikes, where an oligonucleotide library is inserted into a molecular spike entry vector, and the vector pool is linearized and in vitro transcribed to generate a pool of molecular RNA spike-ins. **b**, Coordinates of molecular spikes in basepairs (bp), with inbuilt UMI in the 5′ or 3′ end. **c**, 5′ molecular spike complexity estimated by fitting a nonlinear asymptotic model (dotted line) to unique spUMI sequences observed as a function of the number of spUMIs sequenced across cells (blue line). **d**, Scatter plot showing error-corrected (hamming distance (HD) 1) Smart-seq3 RNA counts (*y* axis) against the number of spiked molecules (*x* axis) ranging from 1 to 100 spiked molecules per cell. Data from HEK293FT cells (*n* = 48 cells). **e**, Scatter plot showing number of spiked molecules (*x* axis) against error-corrected RNA counts (hamming distance 1) for data generated with variations to the Smart-seq3 protocol, that utilize cDNA cleanup before amplification (0.1 μM FWD) or without cleanup and therefore remaining TSO with different concentrations of FWD primer. Data from 39 cells or more are shown per condition. **f**, Scatter plot showing number of spiked molecules (*x* axis) against error-corrected RNA counts (hamming distance 1) for 10x Genomics (v.2) data (*n* = 955 cells). **g**, Scatter plot showing number of spiked molecules (*x* axis) against error-corrected RNA counts (hamming distance 1) for data generated with variations to the SCRB-seq and tSCRB-seq protocols. Standard SCRB-seq (green, 53 cells), excluding exonuclease I treatment (red, 77 cells) and direct PCR (tSCRB-seq) (blue, 90 cells). **h**, Percent counting error (observed/true) for in RNA counts generated with variations to the SCRB-seq and tSCRB-seq protocols. Solid line denotes the mean over cells per condition with the shaded area representing the standard deviation colored by experimental conditions. Direct PCR (tSCRB-seq) (90 cells), No exonuclease I (77 cells) and standard protocol (53 cells). The dotted line represents the expected overcounting if every sequenced read corresponds to a new UMI observation.

preamplification to cause artificially inflated RNA counts (Extended Data Fig. 2d). Whereas the TSO can be removed efficiently by bead cleanups before PCR, it can also be outcompeted effectively by increasing concentrations of forward PCR primers ("FWD"; Fig. 1e). However, the combination of remaining TSO with lower amounts of forward PCR primer results in notable TSO priming

and inflation in RNA counting, at approximately 150% of the correct expression levels (Fig. 1e and Extended Data Fig. 3). We note that a minor count inflation (approximately 110%) is detectable even at 0.5 μM forward primer (at 100 RNA copies per cell and over ten sequencing reads per molecule), and that an increase to 1.0 μM in Smart-seq3 effectively removes this remaining inflation.

Most scRNA-seq protocols rely on 3′-tagging mRNA instead of producing full-length coverage of transcripts, and we therefore engineered a 3′-molecular spike carrying the 18 nt spUMI close to the poly-A tail (Fig. 1b). After similar QC and filtering of 3′ spUMIs (Extended Data Fig. 4), we first applied these molecular spikes to the droplet-generation process in a 10x Genomics Gene Expression Assay (v.2 chemistry; Methods). The inferred molecule counts from this experiment were in good agreement with the molecular spikes (Fig. 1f), as expected since the 10x Genomics protocol purifies the complementary DNA extensively before PCR amplification. Next, we applied the molecular spikes to the single-cell RNA barcoding and sequencing (SCRB-seq) protocol[10]—a plate-based 3′-tagging method that includes cDNA cleanup before cDNA amplification. The RNA counting in SCRB-seq was accurate (Fig. 1g). Recently, T-cell SCRB-seq (tSCRB-seq)[11] was introduced and reported to have greatly increased sensitivity compared with SCRB-seq. In tSCRB-seq, the PCR reagents are added directly to the individual reactions without cDNA cleanup (Extended Data Fig. 2e). To assess how RNA counting was impacted in tSCRB-seq, we first generated a SCRB-seq library where we omitted the exonuclease I digest after reverse transcription, which is a safeguard against remaining oligo-dT primer potentially producing faulty amplicons in the subsequent PCR reaction, which resulted in minimal (105%) UMI counting inflation (Fig. 1g). Following tSCRB-seq, we added PCR master mix directly into the individual wells of cDNA product and this 'direct PCR' condition resulted in substantial UMI overcounting (Fig. 1g). In fact, the direct PCR implementation in tSCRB-seq introduced new UMIs in nearly every new sequenced read, resulting in overcounting that linearly follows sequencing depth irrespective of expression level (Fig. 1h). Clearly, the UMI-containing oligo-dT primer seems to be preferentially priming in the pre-amplification PCR reaction, introducing false new UMIs in every cycle. The cleanup after pooling RT products, even in the absence of the exonuclease I digest, seemed to be very efficient at removing the oligo-dT primer. Thus, the reported improvement in sensitivity in tSCRB-seq[11] is completely artificial since the reported increased UMI counts do not correspond to RNA molecules.

**Evaluating computational UMI correction using molecular spikes.** Having demonstrated the important role of the molecular spikes in assessing the RNA counting abilities of scRNA-seq methods, we next systematically investigated UMI error-correction procedures and compared their inference with the ground-truth number of spiked-in molecules. We based this analysis on the experiment with 10x Genomics using 3′-molecular spikes, and we sampled molecular spikes and their associated sequence reads (one to ten reads each) matching 60 equally spaced expression levels between 1 and 1,000 molecules (Extended Data Fig. 5a). Moreover, we directly investigated the effect of the UMI length on error-correction by performing these analyses in parallel on in silico trimmed versions of the observed 10 nt 10x Genomics UMI. Basing the RNA counts on uncorrected UMI observations inflated the counts with increasing inflation in longer UMIs (Fig. 2a,b) reflecting the fact that longer UMI sequences have a higher risks of being affected by PCR and sequencing errors. As expected, the inflated counts increased

also with increasing read coverage and expression levels (Extended Data Fig. 5b). Reassuringly, applying UMI error corrections that collapse UMI observations within a hamming distance of 1 nt (as implemented in the zUMIs pipeline[8]) removed a large proportion of counting errors for the longer UMI lengths (Fig. 2c,d) and fully removed the dependency on coverage (Extended Data Fig. 5c). In contrast to a previous report[12], we observe that UMIs of a length of 6 nt or lower had elevated collision rates leading to undercounting even before applying UMI error corrections that led to further reductions in counts (Fig. 2a,b). While empirical Bayes correction algorithms have been proposed to account for the lower coding capacity, their run time is prohibitive for larger datasets[9]. Moreover, only UMI lengths of 8 nt or higher counted RNAs accurately over the full spectrum of assessed expression levels (Fig. 2c,d).

Many common scRNA-seq pipelines have implemented UMI error corrections at an edit distance of 1 nt, and we next compared the RNA counting accuracy by collapsing the same data using edit distances of 1 and 2 nt and compared the counts to the ground-truth based on the spiked-in molecules. While a hamming distance of 1 nt was clearly more suitable for UMIs of length 8 nt, allowing up to two mismatches in 10 nt UMIs improved RNA counting throughout the full range of expression levels (Fig. 2e,f). Finally, we compared several computational strategies that collapse UMIs based on their edit distances and frequencies of observations[7] (Extended Data Fig. 6) and compared their inferred counts to the ground-truth spiked-in molecules. Differences among the collapsing strategies were apparent only for UMIs of 8 nt in length (Fig. 2g,h), where the aggressive collapsing strategies ('cluster' and 'adjacency') underestimate RNA counts due to the collapsing of several molecules at higher expression levels, likely due to coding space exhaustion. In line with previous findings[7], the 'directional-adjacency' method seems to provide a good compromise for UMIs of at least 8 nt.

**Complex set of molecular spikes.** RNA spike-in pools (for example, ERCCs[13], SIRVs[14] and Sequins[15]) have been used to correlate estimated spike-in molarities to observed counts and to normalize endogenous RNA counts[16]. The ability to count molecular spikes experimentally (via the internal spUMIs) allows for more accurate experiments and introduces the ability to detect both computational and experimental problems. To this end, we designed a highly complex set of 264 molecular spikes, based on 11 unique spike sequences spanning different lengths (570–3,070 nt) and GC contents (40–60%) (Fig. 3a and Table 1). To precisely evaluate quantification over different expression levels, transcript lengths and GC contents, we cloned 7-nt barcodes (BCs) in twofold abundance steps into each spike sequence (12 steps in duplicates; 24 barcodes per sequence) creating a standard curve for each spike sequence (Fig. 3a and Extended Data Fig. 8). To determine the molecular abundance of each of the 264 molecular spike-ins (the 'ground truth'), we performed an exhaustive sequencing across the spike barcode and spUMIs (here, 14N) and determined the total complexity in the pool to be 76 million unique molecules (Extended Data Fig. 7a,b).

Next, we added the set of molecular spikes to individual HEK293FT cells generated with Smart-seq3xpress protocol[17]. Individual HEK cells were dispensed using an F.SIGHT OMICS

**Fig. 2 | Evaluation of computational RNA counting strategies using molecular spikes. a–d**, Counting difference between number of unique spike identifiers and quantified 10x Genomics UMIs at varying mean expression levels. Colored lines indicate the mean ($n = 100$ in silico cells) counting difference per UMI length shaded by the standard deviation. Counting difference is expressed in absolute numbers (**a,c**) or as a percentage of the mean spUMI count (**b,d**), and UMI counts were computed without error-correction (**a,b**) or corrected in adjacency mode (hamming distance 1) (**c,d**). **e,f**, Comparison of edit distance (hamming distance) for adjacency error correction of UMIs of length 8 or 10 nt. Lines indicate the mean ($n = 100$ in silico cells) difference in quantification between spUMIs and UMIs shaded by the standard deviation in absolute scale (**e**) or relative to the mean (**f**). **g,h**, Evaluation of computational UMI collapse methods adjacency, adjacency-singleton, adjacency-direction and cluster at edit distance 1 and UMI lengths of 8 or 10 nt. Lines indicate the mean ($n = 100$ in silico cells) difference in quantification between spUMIs and UMIs shaded by the standard deviation in absolute scale (**g**) or as a percentage relative to the mean (**h**).

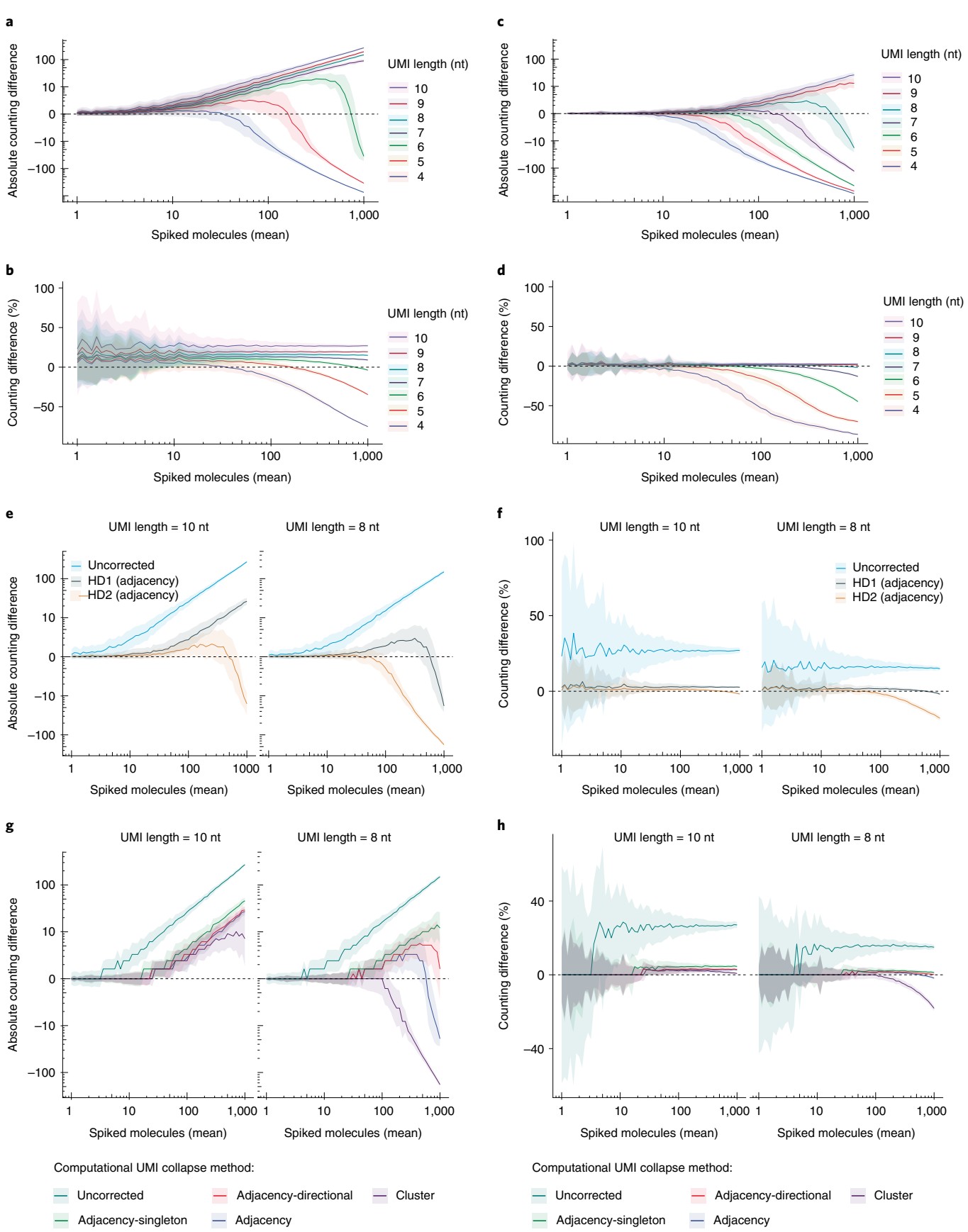

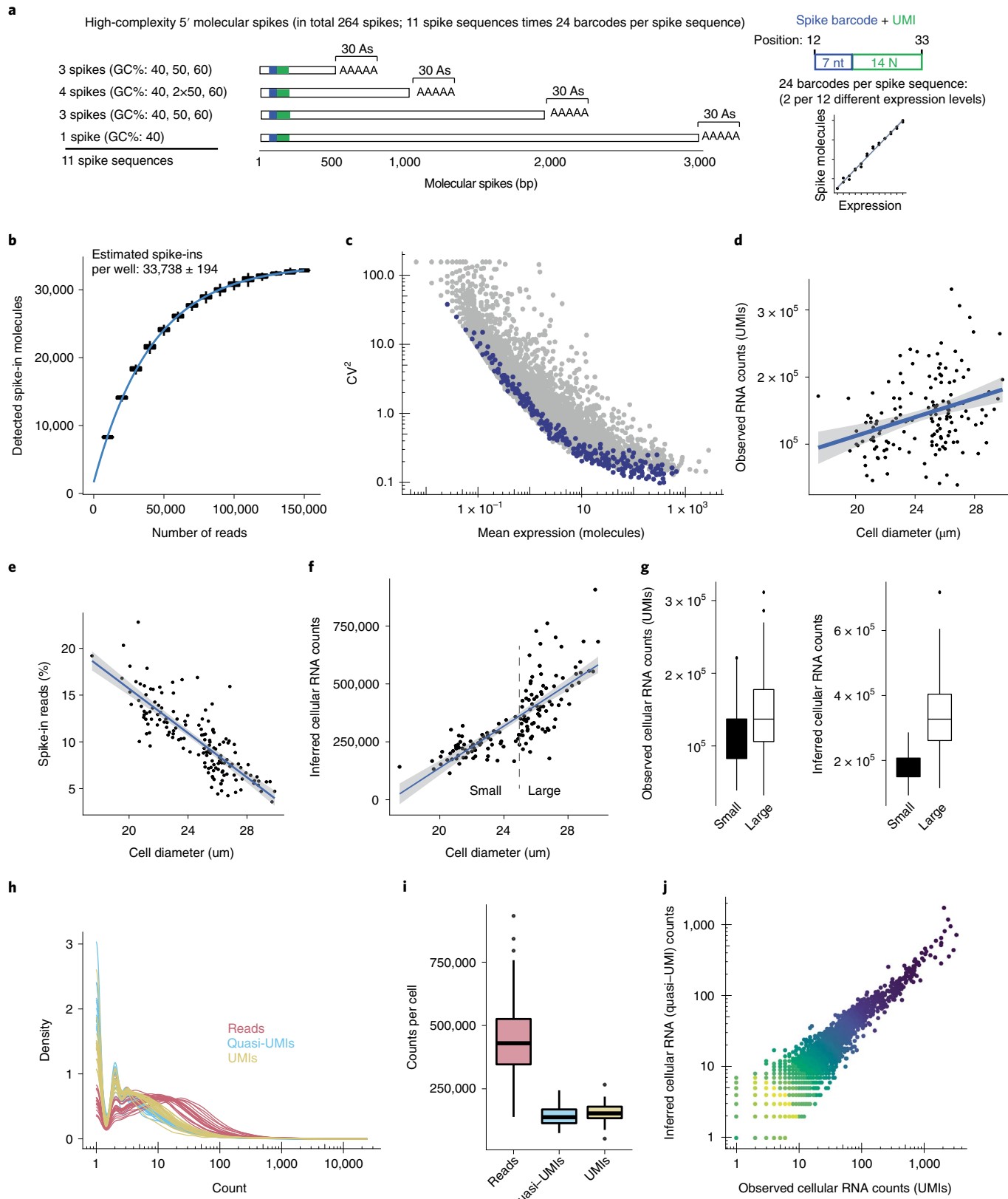

(Cytena) into individual wells while recording cell diameters. Libraries from control wells lacking cells (therefore containing essentially only molecular spikes; Extended Data Fig. 7d) were used to quantify the numbers of spike-in molecules dispensed per well (on average 33,378; Fig. 3b). Contrasting the mean-variance

relationship for endogenous genes and the 264 spike-ins confirmed that the set of molecular spikes spanned relevant endogenous expression levels and that they accurately modeled technical variation in the homogenous HEK cell population (Fig. 3c). We inferred the spike-in detection sensitivity of Smart-seq3xpress to

**Fig. 3 | Single-cell RNA counting using the complex set of molecular spike-ins. a**, Illustration of the design of the complex set of molecular spike-ins. The set consists of 264 unique spike-ins based on 11 distinct spike sequences of different lengths and GC levels, as shown. Each of these 11 sequences further contains 24 different barcodes that were introduced in titration to obtain a standard curve per spike-in sequence (using 2 barcodes per expression level, covering 12 different expression levels). **b**, Boxplot showing the number of spike-in molecules detected per well as a function of sequence depth, only using the wells lacking cells ($n = 942$). **c**, Scatter plot showing the mean molecules detected per cell ($x$ axis) against the squared CV ($y$ axis) over HEK293FT cells ($n = 151$). The 264 molecular spikes are colored in blue, while endogenous genes ($n = 32,738$) are colored gray. **d**, Scatter plot showing the observed number of RNA counts of cellular genes (UMIs) against cell diameter, which was recorded while dispensing each cell into wells. Linear regression shown with line and 95% confidence interval (gray shaded area). **e**, Scatter plot of percent reads aligning to spike-ins per cell against the recorded cell diameter. Linear regression shown as line and 95% confidence interval (gray shaded area). **f**, Scatter plot showing inferred cellular RNA counts against recorded cell diameter, with linear regression shown as line and 95% confidence interval (gray shaded area). **g**, Boxplots showing observed (left) and inferred (right) cellular RNAs for large ($>25\,\mu m$) and small ($<25\,\mu m$) HEK cells ($n = 84$ and 67, respectively). **h**, Distributions of nonzero reads, inferred quasi-UMIs and observed Smart-seq3 UMIs across genes, shown for 20 representative HEK cells. **i**, Boxplots showing the total number of reads, inferred quasi-UMIs and observed Smart-seq3 UMIs per cell ($n = 151$). **j**, Scatter plot of observed Smart-seq3 UMIs against inferred quasi-UMIs across all genes ($n = 17,054$) for a representative cell. The boxplots shown in **g** and **i** show the median, first and third quartiles as a box, and the whiskers indicate the most extreme datapoint within 1.5 lengths of the box.

**Table 1 | Summary statistics of the complex 5′ molecular spikes set**

| Name | Barcodes | Length (nt) | GC content (%) | Total complexity (molecules) | Range (molecules per BC) |
|---|---|---|---|---|---|
| Molecular spike 1 | 24 | 2,070 | 58.5 | 1,746,869 | 205–405,467 |
| Molecular spike 2 | | 1,070 | 58.7 | 4,628,714 | 789–1,058,215 |
| Molecular spike 3 | | 570 | 54.7 | 8,735,160 | 2,726–1,815,444 |
| Molecular spike 4 | | 2,070 | 48.2 | 9,203,773 | 2,823–1,812,543 |
| Molecular spike 5 | | 1,070 | 49.3 | 12,828,533 | 3,649–4,337,579 |
| Molecular spike 6 | | 570 | 44.2 | 9,079,194 | 4,941–1,465,458 |
| Molecular spike 7 | | 3,070 | 39.3 | 4,399,937 | 1,047–1,029,909 |
| Molecular spike 8 | | 2,070 | 38.8 | 7,130,428 | 3,146–1,416,569 |
| Molecular spike 9 | | 1,067 | 39.4 | 5,790,596 | 1,771–1,169,341 |
| Molecular spike 10 | | 568 | 40.0 | 8,121,606 | 5,032–1,395,030 |
| Molecular spike 11 | | 1,096 | 49.1 | 4,946,866 | 666–1,033,909 |

68% (Methods) based on observing, in each well, 33,378 of the expected 49,125 spike-in molecules (corresponding to 0.0321 pg of spike-ins), and abundances were highly correlated to spUMI counts (Extended Data Fig. 7c; Spearman's $r = 0.99$).

We then investigated to what extent the total detected RNAs per cell followed the cell diameter, which showed only modest correlation (Fig. 3d; Spearman's $r = 0.34$), whereas total RNAs detected per cell strongly correlated with the number of sequenced reads (Extended Data Fig. 7e). The relative number of sequenced reads originating from spike-ins and endogenous mRNAs was clearly anticorrelated (Spearman's $r = -0.83$) with cell diameter (Fig. 3e and Extended Data Fig. 7f). This indicates that the mRNA content of cells scales with cell diameters which is masked by varying numbers of reads per cell and thus differing levels of sequencing saturation. Using the quantified ratio of spike-in molecules to endogenous molecules, with the absolute number of captured spike-in molecules per well, we inferred total mRNA content per HEK cell that correlated well with cell diameter (Fig. 3f; Spearman's $r = 0.83$), and ranged between ~250,000 and ~400,000 poly-A+ mRNAs, for smaller and larger HEK293FT cells, respectively (Fig. 3g).

We next explored whether the set of molecular spike-ins could be used to infer gene-level RNA counts in experiments with inflated UMI-based counts or even in experiments lacking UMI-based counting altogether. Recently, a quantile normalization procedure was shown to effectively transform read-count data to molecule counts[18]. That approach, however, requires a shape parameter of the target RNA count distribution to be provided from already existing data of the same cell type. We realized that the RNA counts obtained

from the internal spUMIs could be used to overcome this limitation, and we provided those to the maximum-likelihood estimate of the Poisson-lognormal shape parameter for every cell. Knowing the ratio of reads aligning to spike-in and endogenous RNAs, we then computed total cellular RNA amounts, which we used to fine-tune the shape parameters so that the sum of inferred molecules (called quasi-UMIs) per cell differs minimally from this estimate.

We applied this correction strategy to the experiment on HEK293FT cells and the raw read counts that we used as an example of 'inflated' RNA counts. The method-inferred quasi-UMIs could then be compared with the RNA counts obtained with the Smart-seq3-based UMI (as the true RNA counts). Visualizing the count distribution of reads, inferred quasi-UMIs and the true UMIs for 20 representative cells (Fig. 3h) revealed that the quasi-UMIs are indeed close to the true UMIs. As expected, the total cellular counts of quasi-UMIs approached the observed UMIs per cell (Fig. 3i) and, importantly, their expression levels correlated strongly over genes (Fig. 3j; Spearman's $r = 0.93$). We conclude that the set of molecular spike-ins correct RNA counts in experiments with inflated counting, even in experiments completely lacking UMIs (for example, relevant for Smart-seq2 (ref. [19])).

## Discussion
Here, we developed molecular spikes, that is, a set of RNA spike-ins that contain an inbuilt UMI (Fig. 1a,b), to detect, quantify and correct artifactual RNA counting. The ability to quantitatively monitor the exact spike-in molecules sequenced from each cell is thus not dependent on the level of accuracy when

distributing spike-ins across cells, since the molecular spikes harbor an internal high-capacity spUMI. The quantitative comparison of spiked molecules to the counted RNA revealed both gross (for example, 400%; Fig. 1g) and smaller (5–10%) counting errors (Fig. 1e,g), both relating to procedures that did not sufficiently remove UMI-containing oligonucleotides from contributing during PCR. We therefore suggest that molecular spikes should be applied widely to scRNA-seq method developments to allow accurate reporting of method performances.

Moreover, we demonstrate how molecular spike-ins can be used to rescue faulty experiments, or even enable counting in experiments lacking UMIs. Another benefit from routine use of molecular spike-ins lies in the ability to infer total mRNA amounts per cells within and across cell types (Fig. 3g). Therefore, widespread inclusion of molecular spike-ins in cell atlas projects would reveal cell-type variation in transcriptome complexity and can reveal how mRNA amounts relate to other cellular properties, exemplified here by cell diameter (Fig. 3f). To this end, we are making the molecular spikes available to academic users along with an R package for molecular spike data processing, quality control, rescue in RNA counting and visualization (https://github.com/sandberg-lab/molecularSpikes).

The generation of ground-truth molecular counts across cells with molecular spikes enables systematic benchmarking of UMI error-correction strategies as one can quantitatively compare estimated RNA counts with the numbers of spiked molecules per cell. We show direct experimental evidence that RNA counting based on uncorrected UMIs overestimates RNA expression, at a level that follows the chance of PCR and sequencing errors in the UMIs (UMI lengths, sequence depth and sequencing technology used). In contrast to recent recommendations based on computational modeling[20], our direct experimental comparison shows that scRNA-seq data processing should include UMI error-correction to avoid systematically overestimating RNA expression levels. The literature provides conflicting recommendations regarding UMI lengths[12,20], and finding the optimal compromise for scRNA-seq applications is not straightforward as longer UMIs can interfere with method sensitivity by decreasing reaction efficiencies (for example, reverse transcription[21,22]) and shorter UMIs have limited coding capacity. We demonstrate that only UMIs of 8 nt or longer have sufficient coding capacity to robustly detect expression levels, even in high RNA content, cultured cells (here, HEK293FT cells), and the use of shorter UMIs should be avoided except in shallow scRNA-seq experiments. Interestingly, none of the correction strategies typically used were fully robust across expression levels and it should be possible to use the quantitative data from the molecular spikes to inform future improved strategies with increasing RNA counting reliability and accuracy. It should be noted that molecular spikes are most helpful in methods where addition of the UMI occurs early in the protocol (for example, during reverse transcription). It will also be interesting to use the molecular spikes beyond the validation of aggregated RNA counts per cell, and to investigate the within-molecule consistency of molecular spike identity and UMIs assigned to each molecule. In particular, an exact one-to-one mapping between sequence reads and original molecules (after UMI error-correction) is important for in silico RNA reconstruction[6] to ensure the correct collapsing of sequences for each individual RNA molecule present in cells. Since the set of molecular spike-ins (Fig. 3a and Table 1) span different lengths and GC levels, they enable further indepth characterization of method counting and performance.

## Online content

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

## Methods

**Molecular spike-in design.** Molecular spike sequences were designed as random sequences and we confirmed them to have minimal overlap to mouse or human genomes using BLAST. Two 500-bp sequences were selected, and entry vectors were created as described below. To minimize levels of in vitro transcription from the 5′ synthetic spike empty vector, we decided to complete the T7 promoter sequence with the random-base containing oligonucleotide. A similar strategy was not possible for the 3′ synthetic spike.

**5′ and 3′ spike entry vector and library cloning.** Geneblocks encoding synthetic RNA sequences and a synthetic poly-A stretch were introduced into the pUC19 backbone as previously described[6]. The resulting molecular spike insert vectors were linearized by digestion with *Xho*I or *Eco*RI for the 5′ and 3′ spike encoding plasmids, respectively. A single-stranded oligonucleotide library (IDT), containing a stretch of 18 random bases, was cloned into the linearized backbone using Gibson Assembly (NEB). The resulting reaction was then electroporated into Lucigen Endura Electrocompetent cells, according to the manufacturer's protocol, and streaked out on large LB-agar plates (LB-lennox recipe). The resulting cultures were recovered from the LB-agar plates and purified using NucleoBond Xtra Maxi Plus columns (Macherey-Nagel).

**Design of the complex set of molecular spike-ins.** Molecular spike sequences were generated by randomly sampling 1 million random sequences of each desired length (500; 1,000; 2,000; 3,000 nt) and GC content (40%, 50%, 60%). The resulting sequences were then filtered by discarding any that had nucleotide stretches of identical bases for longer than four consecutive bases. We further evaluated the sampled GC content and required it to be within 1% of the desired value. Next, we generated synthetic sequencing reads of 50 bp at 100× coverage for each candidate sequence using the polyester package[23]. Simulated reads were mapped against the human genome (hg38) using STAR[24]. Only candidate sequences without any mapping reads were considered further. Next, we analyzed GC content in a sliding window and ranked remaining candidate sequences by the least local variation in GC content. For the 7-ntide expression-level barcodes, we generated a candidate barcode sequence set with minimal hamming distance of 3 nt to each other barcode using the DNABarcodes R package[25]. Self-complimentary sequences and homonucleotides of more than three consecutive bases were discarded. We then ranked the candidate barcodes by their deviation from 50% GC content to choose the final set of barcodes.

**Cloning of the complex set of molecular spike-ins.** Plasmids containing the sequences of the selected molecular spike sequences were ordered from Genscript, with an incomplete T7 promoter sequence upstream and a hardcoded poly-A inserted downstream in the pUC19 backbone. The plasmids were linearized by digestion with *Xho*I overnight and gel purified. Individual oligos encoding cloning overlaps, T7 promoter-completing sequence, a 9-base spacing sequence, 7-base barcodes, and a 14N UMI sequence were ordered from IDT DNA and pooled at the appropriate relative concentrations using the Agilent Bravo liquid-handling platform. Oligonucleotide sequences are listed in Supplementary Table 1. Gibson assembly (NEB) reactions were performed as per the manufacturer's instructions, and the resulting reactions were desalinated before electroporation using 0.025 μm pore size Millipore filters (catalog no. VSWP01300) to avoid arcing. The entire Gibson assembly mixture was then transformed into Lucigen Endura Electrocompetent bacteria (four aliquots per Gibson reaction) as per the manufacturer's instructions and pooled and grown in 250 ml LB with ampicillin. The resulting bacteria cultures were then purified individually using the NucleoBond Xtra Maxi Plus kit from Macherey-Nagel.

**In vitro transcription reactions.** The plasmid libraries were linearized by digesting with *Hin*dIII or *Nsi*I for the initial and complex set of molecular spikes, respectively. In vitro transcription was performed using the MaxiScript kit (Invitrogen) according to the manufacturer's guidelines. Resulting libraries of synthetic RNA spikes were cleaned up using RNeasy spin columns (Qiagen). Synthetic RNA integrity was confirmed by RNA Nano 6000 chip on the Agilent Bioanalyzer. For the complex set of molecular spikes, the in vitro transcription proved repeatedly inefficient for two (of the three) 3,070 base transcripts, with relatively high-abundance of transcripts of shorter lengths. These were therefore left out of the final set of molecular spikes.

**Exhaustive sequencing of true abundances of the set molecular spike-ins.** A total of 384 replicates of 10 ng spike-in RNAs were each reverse transcribed using the Smart-seq3 protocol at 1:10 and preamplified for eight cycles of PCR. The resulting cDNA was pooled and cleaned using 0.8× 22% PEG beads. Next, we amplified short 100–150 bp amplicons around the transcription start site using the TSO-specific Smart-seq3 FWD primer containing the Nextera Read 1 overhang and a sequence-specific reverse primer for each of the 11 distinct spike-in sequences with a TruSeq Read 2 overhang. Short amplicons were cleaned with 1.8× ratio of 22% PEG beads and single-indexed for eight cycles using a 2× Phusion HF MasterMix (Thermo Fisher). The libraries were then converted to circular ssDNA using the Universal Library Conversion Kit App-A

(MGI). We used 60 fmol of ssDNA for DNA nanoball generation and subsequent sequencing on two FCL flow-cells of the DNBSEQ G400RS platform (MGI; v.1.1.0.108) generating single-end 100 bp reads to a depth of approximately 3,800 million raw reads.

**Cell culture.** HEK293FT cells (Thermo Fisher) were grown in complete DMEM medium supplemented with 4.5 g l⁻¹ glucose, 6 mM L-glutamine, 0.1 mM MEM nonessential amino acids, 1 mM sodium pyruvate, 100 μg ml⁻¹ penicillin-streptomycin and 10% fetal bovine serum (FBS). Before scRNA-seq experiments, cells were dissociated using TrypLE. HEK293FT cells were not recently authenticated.

**10x Genomics library preparation.** We added 1 μl of 3′ molecular spike pool (1 ng μl⁻¹) to the single-cell HEK293FT suspension immediately before loading on the 10x Genomics 3′ V2 chip. To avoid obtaining too many cells, and to remove the possibility of many 'empty' droplets that reverse transcribed only the molecular spike molecules, we opted to remove GEMs from the reaction before recovery of the cDNA. Before adding recovery agent, 10 μl of GEM-RT mix was transferred and the remainder of the GEM-RT mix was discarded. PCR amplification was performed according to the manufacturer's protocol. After PCR amplification, we performed cleanup with SPRIselect beads at a ratio of 0.8:1 beads:sample instead of the 0.6:1 ratio specified in the protocol. The subsequent fragmentation step was extended to 10 min. The double-sided bead-cleanup after the fragmentation was changed to a ratio of 0.6:1 and 1:1, respectively. Similarly, the postligation cleanup (step 3.4) was increased to 1:1 ratio instead of 0.8:1. The double-sided postindexing PCR cleanup was performed at 0.6:1 and 1:1 ratios, respectively. The library was converted to circular ssDNA using the Universal Library Conversion Kit App-A (MGI). We used 60 fmol of ssDNA for DNA nanoball generation and subsequent sequencing on a FCL flow-cell of the DNBSEQ G400RS platform (MGI; v.1.1.0.108) generating 26×150 bp reads.

**Smart-seq3 library preparation.** Single HEK293FT cells were sorted in 384-well plates containing 3 μl Smart-seq3 lysis buffer on a BD FACSMelody sorter with 100 μm nozzle (BD FACSChorus Software v.1.3). After sorting, plates were quickly spun down before storage at −80 °C. We prepared a Smart-seq3 library according to a published protocol[6] with the following modifications. The 3 μl Smart-seq3 lysis buffer per well contained 0.025 pg 5′ molecular spikes. After reverse transcription, each well containing 4 μl of cDNA was cleaned up with 3 μl home-made 22% PEG beads and eluted in 5 μl Tris-HCl pH 8. PCR mix was added as 5 μl to each well, either with or without the addition of TSO. The reaction concentrations for the PCR in 10 μl were as follows: 1× KAPA HiFi Hot-Start PCR mix, 0.3 mM of each dNTP, 0.5 mM MgCl₂, 0 μM, 0.1 μM, 0.5 μM or 1.0 μM forward primer and 0.1 μM reverse primer. In the samples where TSO was added back into the PCR mix, it was done so at 0.8 μM.

**SCRB-seq library preparation.** Single cells were sorted into 96 wells containing 5 μl of lysis buffer (1:500 dilution of 5× Phusion HF Buffer) containing 0.025 pg of 3′ molecular spike pool using a BD FACSMelody sorter with 100 μm nozzle, and frozen at −80 °C. After thawing, lysis was aided by Proteinase K digestion (1 μl of 1:20 diluted Proteinase K (Ambion)) for 15 min at 50 °C. Proteinase K was denatured, and RNA was desiccated by incubation at 95 °C for 10 min after unsealing the plate. Reverse transcription was performed in a volume of 2 μl per well (1 μM barcoded oligo-dT E3V6NEXT Biotin-ACACTCTTTCCCTACACGACGCTCTTCCGATCT[BC6][UMI10] [T30]VN, 1× Maxima RT Buffer, 0.1 mM dNTPs, 1 μM TSO E5V6NEXT ACACTCTTTCCCTACACGACGCrGrGrG and 25 U Maxima H Minus reverse transcriptase) for 90 min at 42 °C. cDNA was pooled and cleaned using SPRI beads and excess primers were digested by incubation with exonuclease I (NEB; 30 min at 37 °C, inactivation 20 min at 80 °C). PCR amplification was performed in 50 μl (0.5 μM SINGV6 primer Biotin-ACACTCTTTCCCTACACGACGC, 1× KAPA HiFi ReadyMix). PCR was cycled as follows: 3 min at 98 °C, 21 cycles of 15 s at 98 °C, 30 s at 67 °C, 4 min at 72 °C and final elongation for 10 min at 72 °C. For the direct PCR condition, we added 3 μl of PCR master mix directly to each well RT product well containing 2 μl of cDNA. Amplified, pooled cDNA was cleaned and quantified. We used 800 pg of cDNA for tagmentation using the Nextera XT kit (Illumina) according to the manufacturer's protocol. The final indexing PCR was performed using a i7 primer and P5NEXTPT5 (AATGATACGGCGACCACCGAGATCTACACTCTTT CCCTACACGACGCTCTTCCG*A*T*C*T*; IDT) to select for correct 3′ fragments. The libraries were pooled and converted to circular ssDNA using the Universal Library Conversion Kit App-A (MGI). We used 60 fmol of ssDNA for DNA nanoball generation and subsequent sequencing on a FCL flow-cell of the DNBSEQ G400RS platform (MGI; v.1.1.0.108) generating 16×150 bp reads.

**Smart-seq3xpress library preparation.** Single HEK293FT cells were sorted in 384-well plates containing 0.3 μl Smart-seq3 lysis buffer on a F.SIGHT OMICS cell dispenser (Cytena) while excluding dead cells (NucGreen 488, Thermo Fisher) and recording the cell diameter for each dispensed cell. After the dispense, plates were quickly spun down before storage in −80 °C. We prepared the Smart-seq3xpress

library according to the published protocol[17] with the following modification: lysis buffer per well contained 0.0321 pg 5′ molecular spike v.2 pool.

**HEK293FT expression levels.** UMI count tables from HEK293FT cells generated using the Smart-seq3 protocol were obtained from ArrayExpression accession E-MTAB-8735. After additional filtering of the cells (minimum number of genes expressed was 7,500, and minimum number of UMIs detected was 50,000), we calculated the mean UMI count for all genes ($n = 10,198$) detected in at least 50% of cells.

**Sequencing data processing.** All sequencing data was processed using zUMIs (v.2.9.5)[8]. Reads with more than three bases below Phred 20 base call scores in the UMI sequence were discarded. Remaining reads were mapped to the human genome hg38 and spike-in references using STAR (v.2.7.3a)[24] and mapped reads were quantified according to Ensembl gene models (Grch38.95) taking into consideration the strand information of the libraries. Error correction of the internal spUMI was applied within each cell barcode using the adjacency algorithm allowing edit distances of 2 nt (hamming distance).

**Computational analysis of individual molecular spike-ins.** All downstream analyses were performed in R (v.4.0.4). Reads aligning to the molecular spike reference sequence were loaded along with the library UMI and barcode information from zUMIs output bam files using Rsamtools[26] and further processed by matching the known sequence upstream and downstream of the internal UMI. Only valid reads that had an 18 nt long internal UMI were considered further.

To investigate the distances of uncorrected, hamming distance 1 nt and hamming distance 2 nt corrected spUMI sequences, we used 5′ molecular spike data generated by the Smart-seq3 protocol and 3′ molecular spike data from the 10x Genomics experiment. For each cell, we calculated all pairwise hamming distances of spUMI sequences in that cell, as well as all pairwise distances to 1,000 randomly sampled spUMI sequences across the whole dataset, using the stringdist package[27].

To estimate the complexity of the molecular spike pool, we counted the number of unique error-corrected spUMI sequences over molecules seen in all cells and fitted a nonlinear asymptotic regression model using the NLSstAsymptotic function, and extracted the asymptote (total complexity) from the coefficients of the model.

Over-represented spike-ins were discarded if they were detected in more than four or eight cells (5′- and 3′-spUMIs, respectively) or with more than 100 raw sequencing reads.

Functions used in the analysis of this manuscript are made available via the UMIcountR package.

**Analysis of counting performance in protocol variations.** For every cell barcode, spUMIs were drawn randomly from all molecular spike molecules in that barcode for 20 expression levels from 1 to 100 molecules. At each expression level and for each cell, we determined the exact number of molecules by drawing from a normal distribution with the given mean and added Poisson noise (s.d. = square root of the mean). All observed sequencing reads associated with each of the drawn molecules were stored and adjacency error correction (hamming distance 1 nt) was applied to the observed UMI sequences derived from the library preparation (for example the UMI in the Smart-seq3 TSO or the UMI in 10x Genomics oligo-dT).

**Evaluation of UMI length and UMI collapse algorithms.** We first selected a pool of eligible spike-in molecules from all cells in the 10x Genomics dataset that fulfilled the following criteria: (1) observed in only one cell barcode and (2) covered with 10–20 sequencing reads. From this pool of 26,815, we sampled molecules at 60 expression levels spaced evenly in log-space from 1 to 1,000 molecules. At each expression level, we sampled the number of spike molecules used for 100 'in silico' cells by drawing from a normal distribution with the given mean and added Poisson noise (s.d. = square root of the mean). All associated sequencing reads were stored, and we shortened the UMI sequence in 1 base increments (3′ to 5′ direction) from ten to four nucleotides. We then applied our R implementations of the following UMI error corrections at each expression level and in silico cell: (1) adjacency: the network of closely related UMI sequences is resolved by collapsing all sequences within the given edit distance (ran with hamming distance 1 and 2 nt in our case) to the most abundant sequence; (2) adjacency-directional: same as adjacency, but the minor nodes can be collapsed only if they have less than 0.5× the reads of the most abundant sequence; (3) adjacency-singleton: same as adjacency, but the minor nodes can be collapsed only if they are observed by exactly one read; (4) cluster: the network of closely related UMI sequences is resolved by collapsing all sequences within the given edit distance to the node with the highest number of read counts. Nodes that were related at the same distance to one of the collapsed sequences and equally or less abundant are then also collapsed to the main node, even if their edit distance is higher than the initial parameter.

**Computational analysis of the set of molecular spike-ins.** All analyses were performed in R (v.4.1.2). Reads aligned to the 11 distinct molecular spikes sequences after processing with zUMIs were loaded and filtered by the following criteria: (1) match to 1 of the 24 expected 7 nt barcode per distinct sequence (with 1 nt edit distance); (2) presence of 14 nt internal spUMI and (3) matching 25 nt

constant sequence downstream of the spUMI (with up to 3 nt edit distance). For the determination of the complexity of the spike-in pool, we used the ground-truth sequencing and computed the unique hamming distance 2 nt corrected spUMI numbers in each of the 264 barcodes. After downsampling to decreasing read depths, we also fit a nonlinear asymptotic model to confirm that all expected spike-in molecules were indeed seen by the ground-truth sequencing.

**Estimation of molecular spike-in amounts per well.** All empty wells, containing spike-ins in the lysis buffer but received no cell (endogenous reads <20% and spike-in mapped reads >80%), were used to infer the captured amount of spUMI molecule counts. For each well, we sampled coverages from 10,000 to 150,000 reads and quantified the number of unique spUMIs over all 264 spike-in transcripts. We then fit a nonlinear asymptotic model to the depth-molecule relationship and derived the asymptote, that is, captured molecules at saturation from the coefficients of the model fit. We compared this estimate of captured molecules to the expectation of molecules added to each well via the lysis buffer. We diluted spike-in RNA from a starting concentration of 10.7 ng µl⁻¹ (determined from three independent Qubit RNA HS measurements, s.d. 6.5%) to an estimated 0.0321 pg µl⁻¹ on average per well. Using the relative molecular abundances of the 11 distinct sequences in the molecular spike transcript mix derived from the ground-truth sequencing experiment, we calculate the exact molecular weight accounting for their distinct lengths and GC contents. Thus, the number of added RNA molecules is, on average, 49,125 with variations expected from Qubit quantification, pipetting accuracy during dilution and dispensing of lysis buffer.

**Quasi-UMI correction procedure.** To account for well-to-well variation in the present amount of molecular spike RNA, we first check the relationship of the observed spike read depth and number of molecules for each given cell. We use the empty-well-derived spike-in detection model to predict whether more or less molecules than expected on average are being detected and correct the expected total amount of spike-in molecules present in the well accordingly. In addition, we can use the model to predict the level of sequencing saturation in the present well. To estimate the cellular RNA content of each cell, we compute the percentage of spike-in reads among mRNA-aligned reads ($nReads_{spike}$ per $nReads_{endogenous} + nReads_{spike}$) and derive the total RNA complexity of the cells from the percentage of spike-in and absolute number of spike-in molecules estimated. Next, we use the count table of spUMI counts per each of the 264 barcodes per cell as input into the maximum-likelihood estimation of the Poisson-lognormal distribution parameters as implemented in the quminorm R package (https://github.com/willtownes/quminorm) v.0.1.0 using the poilog_mle function. We then use the quminorm function to perform the quantile normalization using the determined shape parameters and in parallel also with small step variations of up to ± 10% of the estimated parameter. After the parallel quantile-normalizations, we empirically choose for each cell the output of the shape parameter where the total counts resemble the calculated expectation most closely.

**Reporting Summary.** Further information on research design is available in the Nature Research Reporting Summary linked to this article.

## Data availability
The raw data files for scRNA-seq experiments and molecular spikes ground-truth sequencing have been deposited in Array Express at European Bioinformatics Institute under accession numbers E-MTAB-10372, E-MTAB-11433 and E-MTAB-11448. Human genome build hg38 fasta files and gene annotation in GTF format (Grch38.95) were obtained from Ensembl.

## Code availability
We are making the code for processing, filtering, quality control and visualization of molecular spikes publicly available as a R package (https://github.com/cziegenhain/UMIcountR).

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

## Acknowledgements
We thank H. and U. Lendahl, as well as B. Schmierer, for valuable discussions and M. Eriksson from the Eukaryotic Single Cell Genomics facility (SciLifeLab, Stockholm) for

help with the 10x Genomics library preparation. This work was supported by an EMBO long-term fellowship (ALTF 673–2017) grant to C.Z., an HFSP long-term fellowship (LT000155/2017-L) grant to G.-J.H. and grants to R.S. from the Swedish Research Council (2017-01062), the Knut and Alice Wallenberg Foundation (2017.0110), the Göran Gustafsson Foundation and the Bert L. and N. Kuggie Vallee Foundation.

## Author contributions

G.-J.H. conceived the idea for the study. G.-J.H. and C.Z. designed and cloned molecular spikes. C.Z., M.H.-J. and G.-J.H. performed scRNA-seq experiments. C.Z. performed analyses and generated figures. R.S., C.Z. and G.-J.H. wrote the manuscript. R.S. supervised the work.

## Funding

## Competing interests

The authors declare no competing interests.

## Additional information

**Extended data** is available for this paper at https://doi.org/10.1038/s41592-022-01446-x.

**Correspondence and requests for materials** should be addressed to Rickard Sandberg.

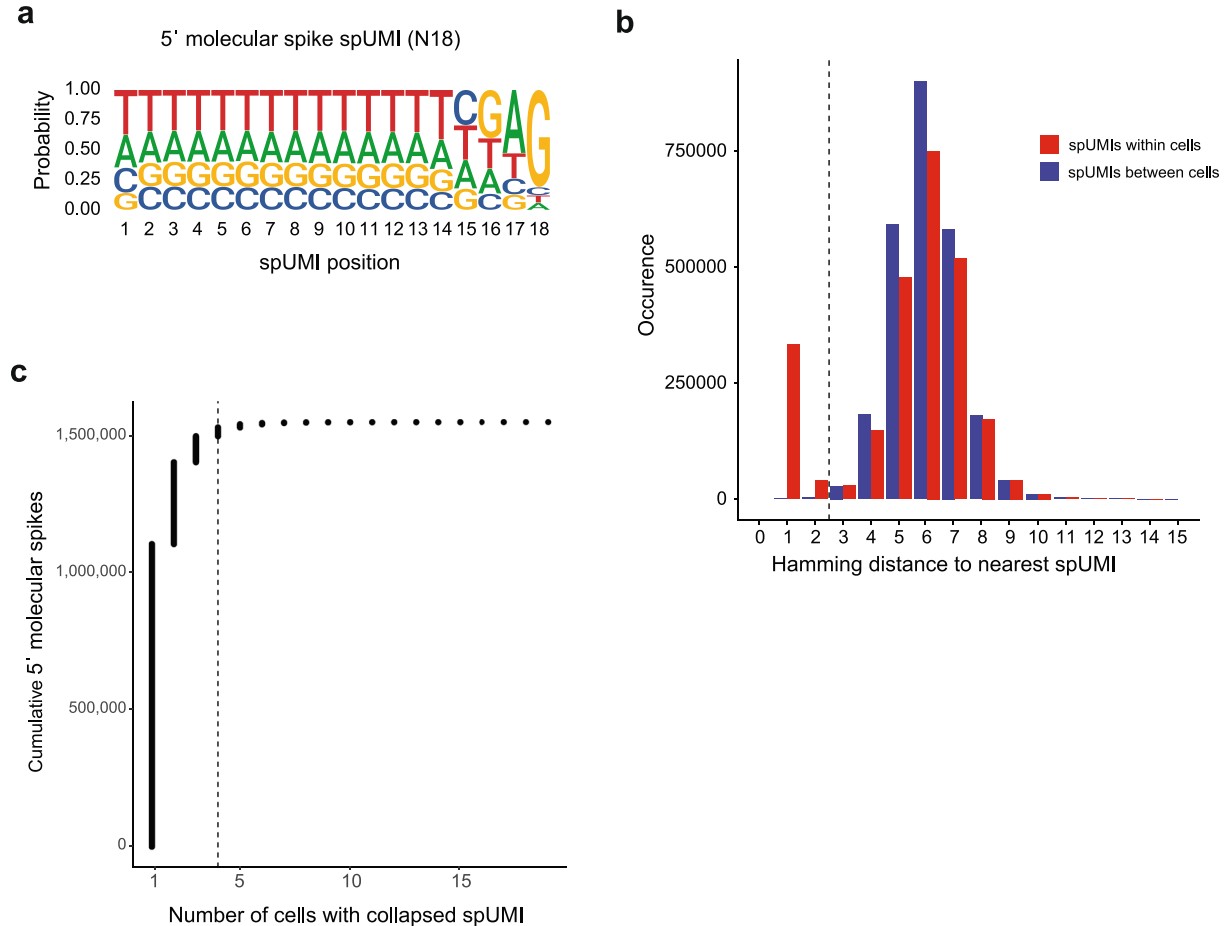

**Extended Data Fig. 1 | Quality control of 5' molecular spike-in.** (**a**) Sequence logo of the 18 random spUMI bases derived from all reads in the Smart-seq3 dataset. At each position, the frequency of all 4 bases is visualized by the size of the DNA letter. (**b**) Minimal distance of uncorrected spUMIs to the closest spUMI sequence for all pairwise within-cell comparisons and pairwise comparisons of spUMIs to 1000 randomly samples spUMI sequences across cells (total 2,233,878 comparisons). (**c**) Cumulative number of molecular spikes (n = 885,925) sorted by their occurrence over cells (n = 340). Dashed line indicates the chosen QC cutoff at 4 cells.

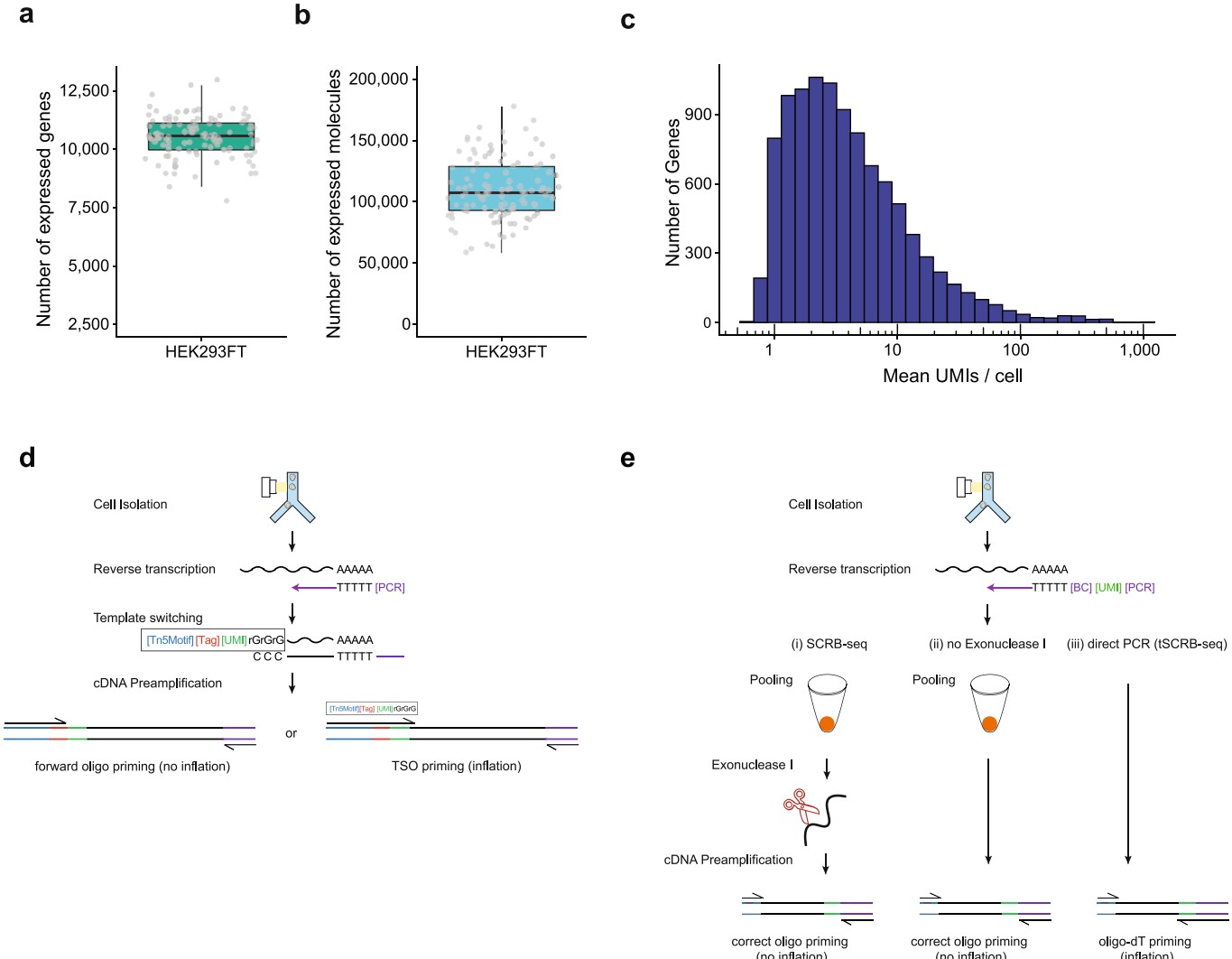

**Extended Data Fig. 2 | Expression levels in HEK293FT cells. (a,b)** Quality of Smart-seq3 libraries (n = 111 cells) after filtering. Shown are the number of detected (a) genes and (b) molecules per HEK293FT cell. Boxplots show the median, first and third quartiles as a box, and the whiskers indicate the most extreme datapoint within 1.5 lengths of the box. (**c**) Histogram showing the mean UMI count per cell for all genes expressed in at least 50% of cells (n = 10,198 genes). (**d**) Schematic overview of the counting-critical steps in the Smart-seq3 protocol. (**e**) Schematic overview of the evaluated protocol variations for the SCRB-seq library preparation.

**a**

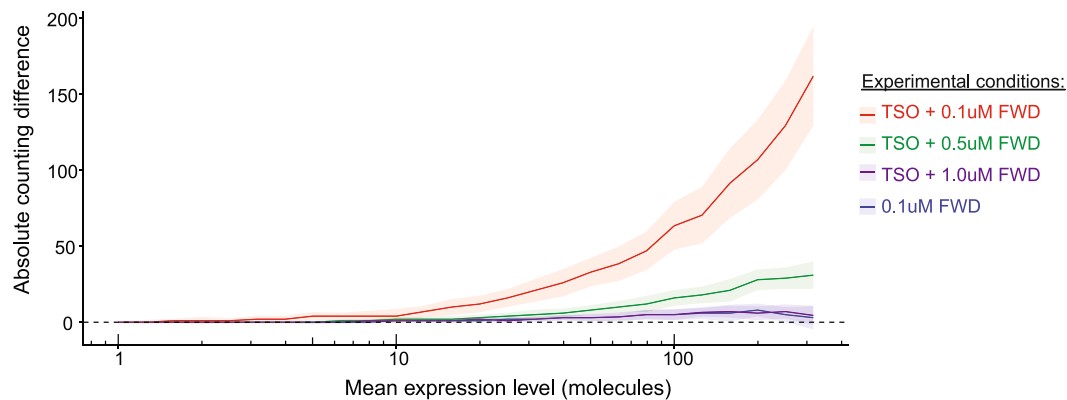

**b**

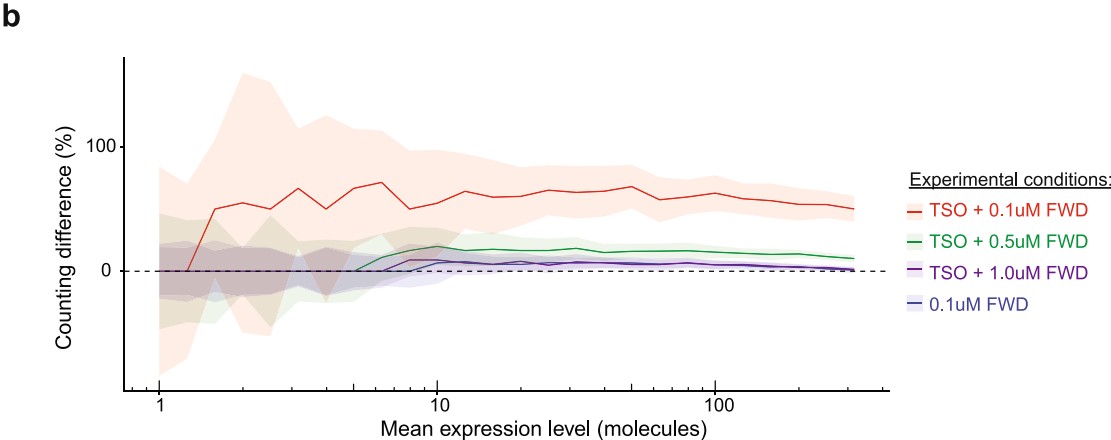

**Extended Data Fig. 3 | Counting difference in Smart-seq3 protocol variations.** (**a**,**b**) For variations of the Smart-seq3 protocol, molecular spikes were sampled at varying mean expression levels. Colored lines indicate the mean counting difference in (a) absolute numbers or (b) relative to the mean and shaded by the standard deviation for library preparation conditions 0.1 μM FWD (n = 48 cells), TSO + 0.1 μM FWD (n = 48 cells), TSO + 0.5 μM FWD (n = 39 cells) and TSO + 1.0 μM FWD (n = 45 cells).

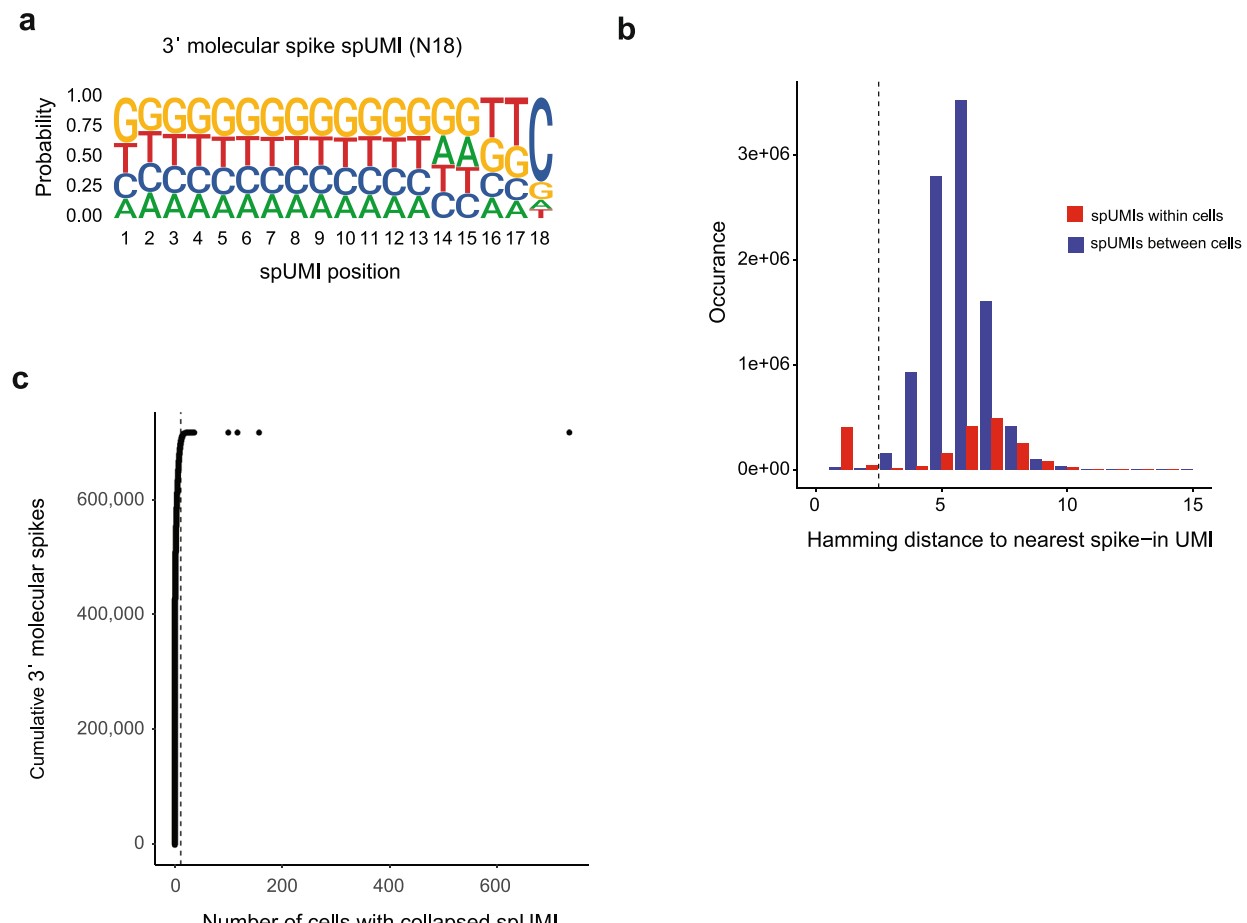

**Extended Data Fig. 4 | Quality control of 3′ molecular spike-in.** (**a**) Sequence logo of the 18 random spUMI bases derived from all reads in the 10x Genomics dataset. At each position, the frequency of all 4 bases is visualized by the size of the DNA letter. (**b**) Minimal distance of uncorrected spUMIs to the closest spUMI sequence for all pairwise within-cell comparisons and pairwise comparisons of spUMIs to 1000 randomly samples spUMI sequences across cells (total 19,773,932 comparisons). (**c**) Cumulative number of molecular spikes (n = 1,938,392) sorted by their occurrence over cells (n = 1,359). Dashed line indicates the chosen QC cutoff at 4 cells.

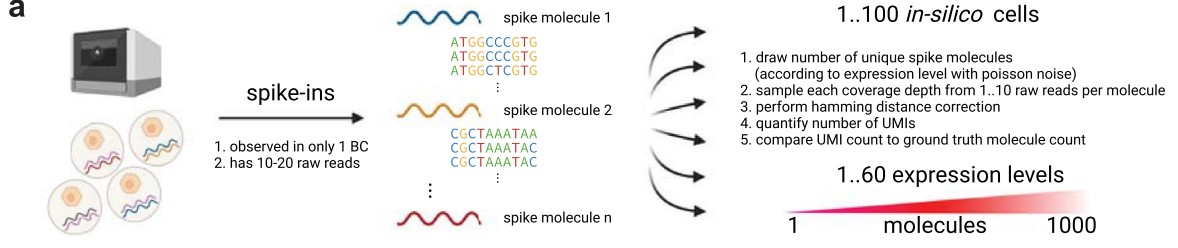

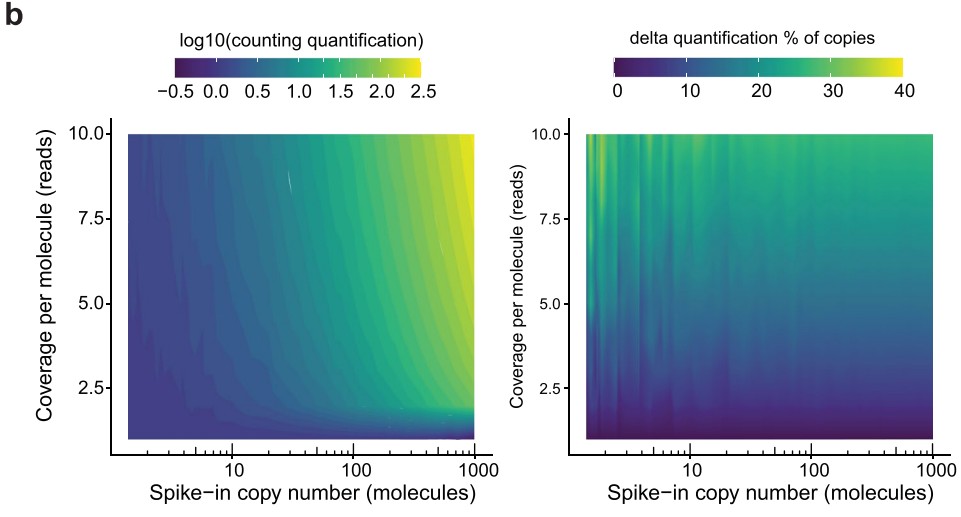

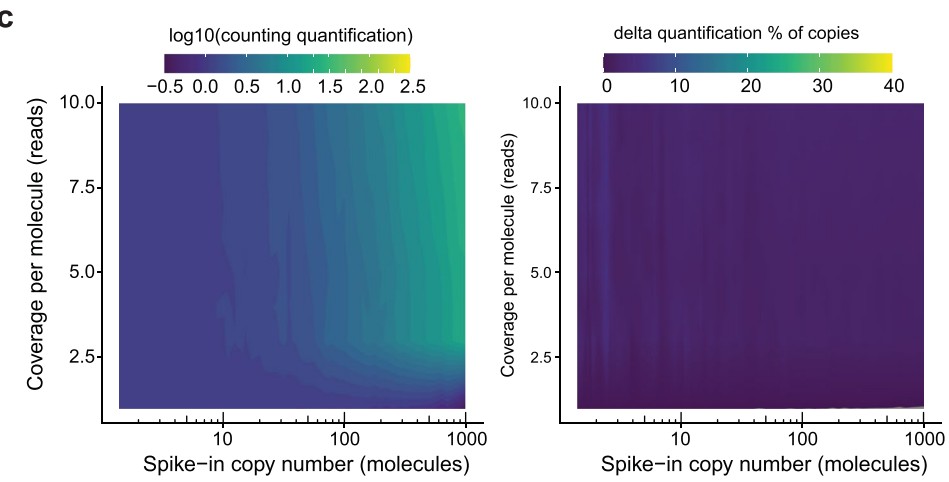

**Extended Data Fig. 5 | Strategy for sub-sampling molecular spikes to assess counting reliability across expression levels. (a)** Strategy for computational analysis of 10x Genomics spUMI data. Molecular spike-ins observed in only one cell barcode and covered by 10–20 sequencing reads are selected along with their associated 10x UMI sequence. spUMIs were sampled at 60 expression levels ranging from 1 to 1000 molecules for 100 *in silico cells*. For each 'cell' at each expression level, molecules were analyzed at depth of 1 to 10 reads and UMI error correction was applied. Created with Biorender.com **(b,c)** We quantified the spUMIs and 10x UMIs and display the mean counting difference over the 100 replicates as a contour plot depending on expression level and read coverage in absolute numbers and normalized to the mean copy number, where (b) shows uncorrected 10x UMI counts and (c) shows UMI counts after applying an error correction at hamming distance 1. In each of the contour plots, the left panel is colored by the deviation from ground truth counting on a log10 scale with a pseudocount of 1 added and the right side denotes the deviation from ground truth relative to the mean expression level.

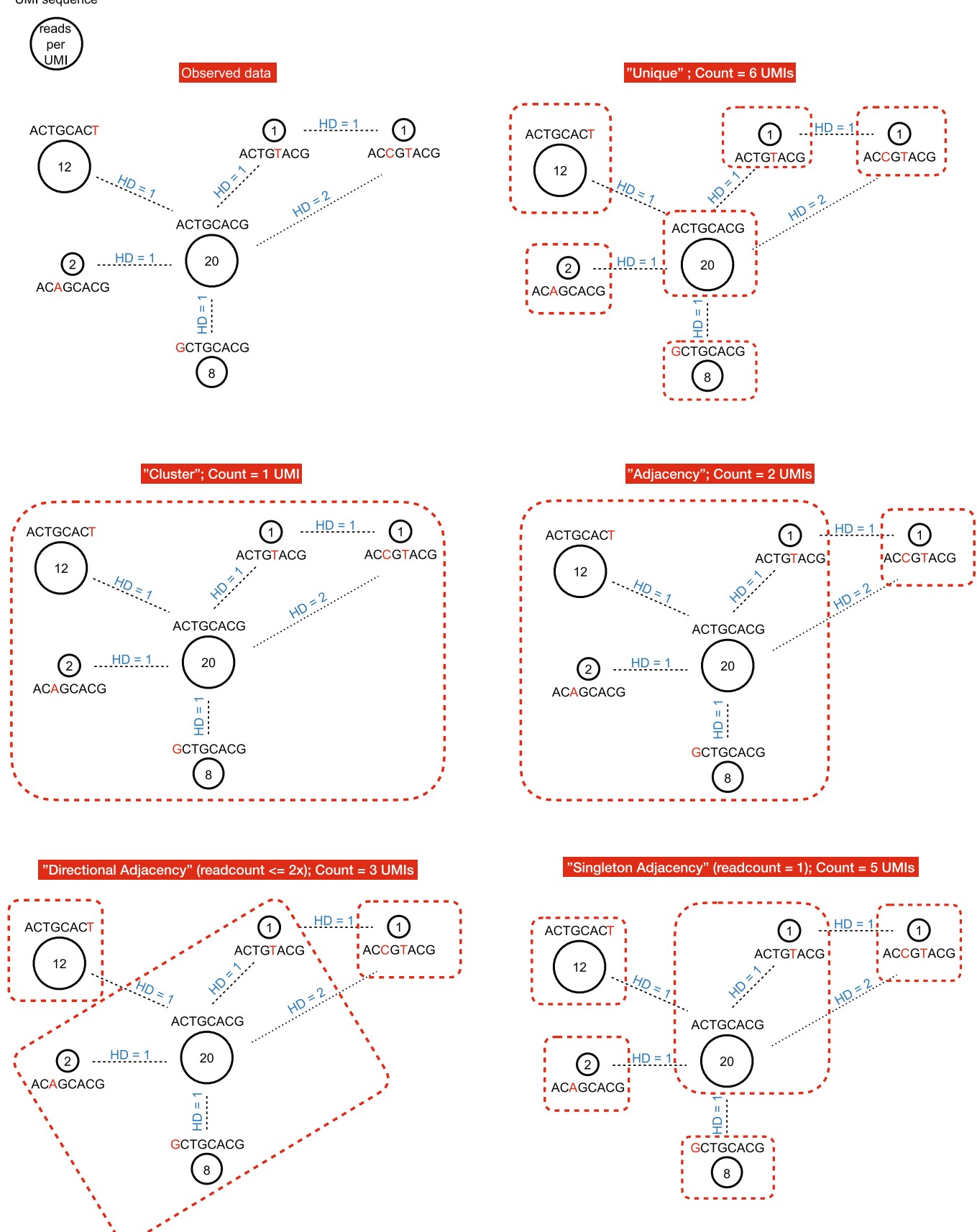

**Extended Data Fig. 6 | See next page for caption.**

**Extended Data Fig. 6 | Computational algorithms for UMI collapsing. (a)** Scenario of a network of UMI sequences where each UMI sequence is visualized along with the number of reads it was observed by. Mismatches to the center UMI sequence are shown in red and the edit distance (hamming distance HD) is indicated in blue. **(b)** Unique: Every unique sequence is counted as a molecule (naive counting, e.g. Kallisto). UMI count in the network = 6. **(c)** Cluster: The network is resolved by collapsing all sequences within HD1 to the UMI with the highest number of read counts. UMIs that were related at HD1 to one of the collapsed sequences and equally or less abundant are then also collapsed to the main UMI sequence, even if their edit distance is higher than 1. UMI count in the network = 1. **(d)** Adjacency: The network is resolved by collapsing all sequences within HD1 to the UMI with the highest number of read counts. UMI count in the network = 2. **(e)** Directional Adjacency: The network is resolved by collapsing all sequences within HD1 to the UMI with the highest number of read counts, unless they are observed with more than 50% of read support compared to the main UMI. UMI count in the network = 3. **(f)** Singleton Adjacency: The network is resolved by collapsing all sequences within HD1 and observed with only 1 read to the UMI with the highest number of read counts. UMI count in the network = 5.

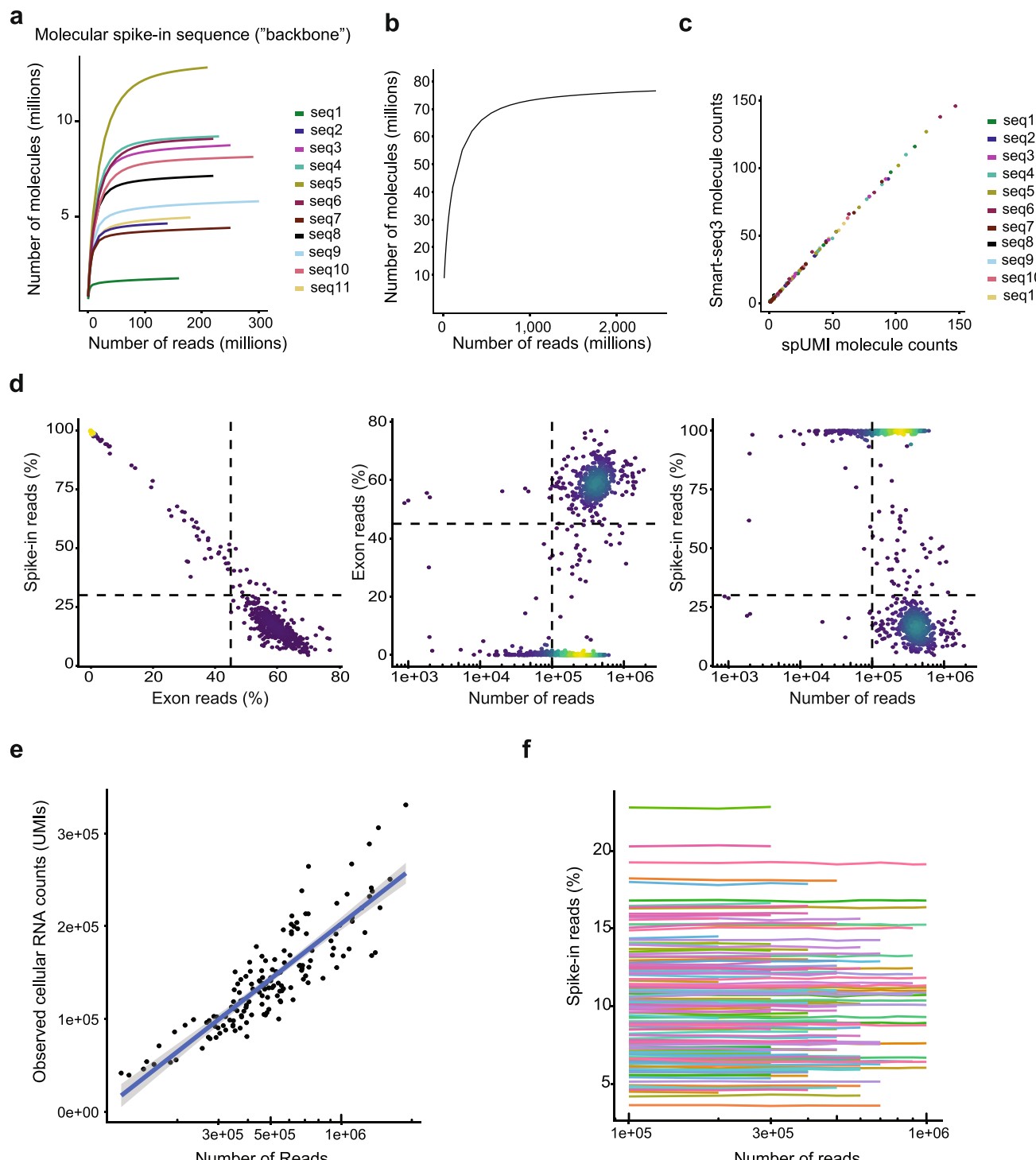

**Extended Data Fig. 7 | Complex set of molecular spike-ins.** (**a,b**) Number of unique spike-in molecules determined in the ground-truth complexity sequencing experiment as a function of sequencing depth for (a) each of the distinct spike-in sequences or (b) overall. (**c**) Scatter plot showing the Smart-seq3 molecule counts (y-axis) versus spUMI molecule counts (x-axis) in a randomly drawn HEK293FT cell. Dots are colored by spike-in transcript sequence. Spearman rank correlation r = 0.99. (**d**) Read mapping statistics for wells with HEK293FT cells and spike-ins only. Left panel: percentage of reads mapping to human exons (x-axis) against percentage of reads mapping to molecular spikes. Middle panel: Percentage of reads mapping to human exons (y-axis) against sequenced reads per well. Right panel: Percentage of reads mapping to molecular spikes (y-axis) against sequenced reads per well. (**e**) Scatter plot showing observed cellular RNA counts (y-axis) against the number of sequenced reads (x-axis), per cell, with linear regression shown as line and 95% confidence interval as gray shaded area. (**d**) The percent of reads aligning to molecular spike-ins as a function of sequence depth, with each colored line showing a unique cell.

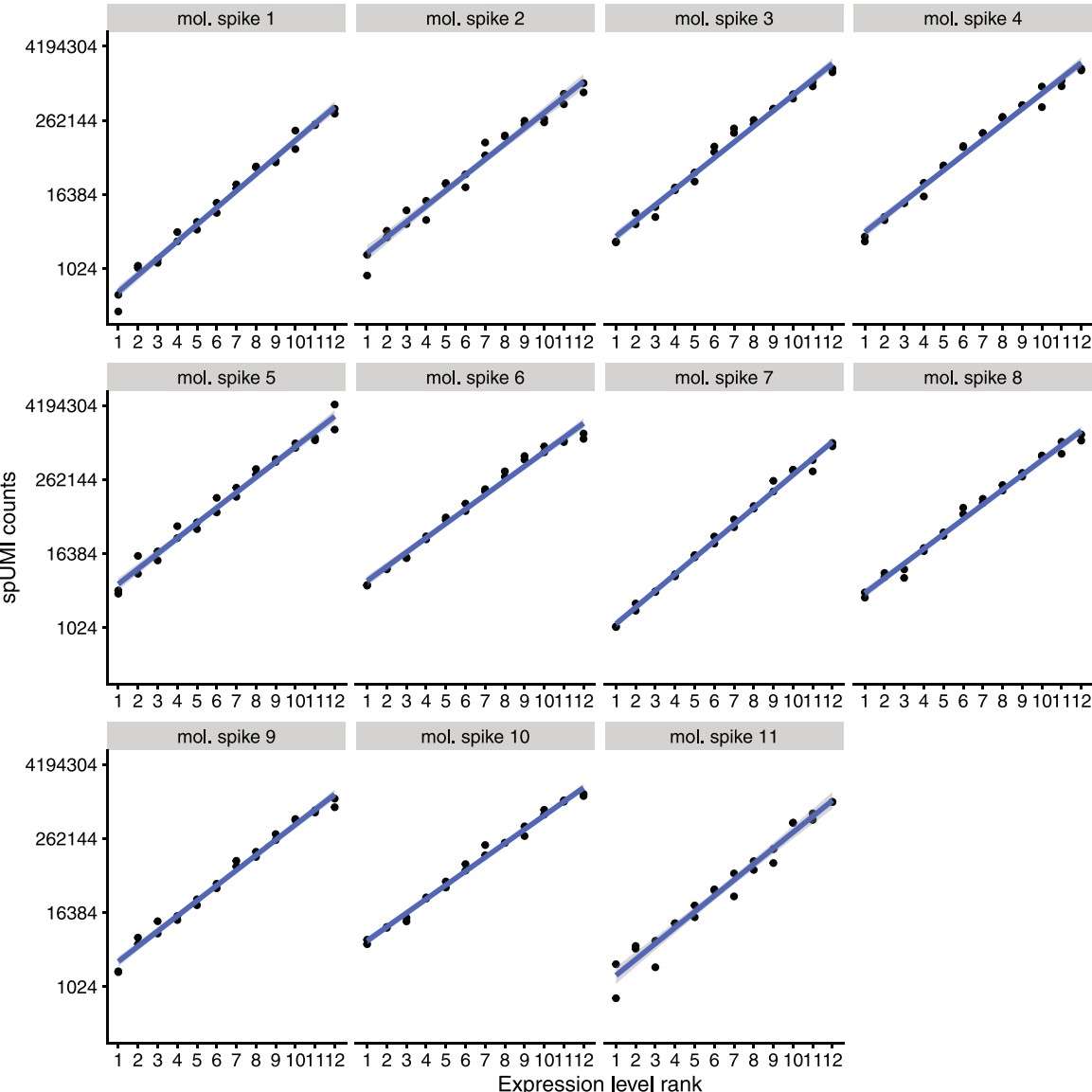

**Extended Data Fig. 8 | Molecular spike internal barcode-based expression standard curves.** For each of the 11 distinct spike-in sequences, we show the total number of unique spike-in molecules per barcode (y-axis; log2 transformed axis) when sequenced to saturation. On the x-axis, we denote the expected expression rank of each barcode. Linear regression shown as line and 95% confidence interval as gray shaded area.

# Reporting Summary

Nature Research wishes to improve the reproducibility of the work that we publish. This form provides structure for consistency and transparency in reporting. For further information on Nature Research policies, see our Editorial Policies and the Editorial Policy Checklist.

## Statistics

For all statistical analyses, confirm that the following items are present in the figure legend, table legend, main text, or Methods section.

| n/a | Confirmed | |
|---|---|---|
| ☐ | ☒ | The exact sample size (*n*) for each experimental group/condition, given as a discrete number and unit of measurement |
| ☐ | ☒ | A statement on whether measurements were taken from distinct samples or whether the same sample was measured repeatedly |
| ☒ | ☐ | The statistical test(s) used AND whether they are one- or two-sided *Only common tests should be described solely by name; describe more complex techniques in the Methods section.* |
| ☒ | ☐ | A description of all covariates tested |
| ☒ | ☐ | A description of any assumptions or corrections, such as tests of normality and adjustment for multiple comparisons |
| ☐ | ☒ | A full description of the statistical parameters including central tendency (e.g. means) or other basic estimates (e.g. regression coefficient) AND variation (e.g. standard deviation) or associated estimates of uncertainty (e.g. confidence intervals) |
| ☒ | ☐ | For null hypothesis testing, the test statistic (e.g. *F*, *t*, *r*) with confidence intervals, effect sizes, degrees of freedom and *P* value noted *Give P values as exact values whenever suitable.* |
| ☒ | ☐ | For Bayesian analysis, information on the choice of priors and Markov chain Monte Carlo settings |
| ☒ | ☐ | For hierarchical and complex designs, identification of the appropriate level for tests and full reporting of outcomes |
| ☒ | ☐ | Estimates of effect sizes (e.g. Cohen's *d*, Pearson's *r*), indicating how they were calculated |

*Our web collection on statistics for biologists contains articles on many of the points above.*

## Software and code

Policy information about availability of computer code

| Data collection | The following software was used in data collection: FACS sorter software (BD FACSChorus 1.3), sequencer software (MGI DNBSEQ G400RS 1.1.0.108). |
|---|---|
| Data analysis | Analysis was performed using zUMIs v2.9.5, STAR v2.7.3a, samtools v1.10, pigz v.2.4, R v4.0.4 or v4.1.2, ggplot2 v3.3.3, data.table v1.14.0, Rsamtools v2.6.0, stringdist v0.9.6.3, UMIcountR v0.1.1 (https://github.com/cziegenhain/UMIcountR) |

For manuscripts utilizing custom algorithms or software that are central to the research but not yet described in published literature, software must be made available to editors and reviewers. We strongly encourage code deposition in a community repository (e.g. GitHub). See the Nature Research guidelines for submitting code & software for further information.

## Data

Policy information about availability of data

All manuscripts must include a data availability statement. This statement should provide the following information, where applicable:
- Accession codes, unique identifiers, or web links for publicly available datasets
- A list of figures that have associated raw data
- A description of any restrictions on data availability

The raw data files for single-cell RNA-sequencing experiments and molecular spikes ground trurth sequencing have been deposited in Array Express at European Bioinformatics Institute under accession E-MTAB-10372, E-MTAB-11433 and E-MTAB-11448. Human genome build hg38 fasta files and gene annotation in GTF format (Grch38.95) were obtained from Ensembl.

# Field-specific reporting

Please select the one below that is the best fit for your research. If you are not sure, read the appropriate sections before making your selection.

☒ Life sciences        ☐ Behavioural & social sciences        ☐ Ecological, evolutionary & environmental sciences

For a reference copy of the document with all sections, see nature.com/documents/nr-reporting-summary-flat.pdf

# Life sciences study design

All studies must disclose on these points even when the disclosure is negative.

| | |
|---|---|
| Sample size | A sample-size calculation was not performed before the experiments. Sample size was maximized within reasonable budget for the cost of preparing libraries and sequencing. |
| Data exclusions | Data filtering steps according to established criteria in single-cell RNA-sequencing were applied only to remove technically failed libraries and described where appropriate. Further data exclusions were not performed. |
| Replication | For each experimental condition, a large number of single cells was sequenced to ensure reprodocibility. Sample sizes are clearly indicated throughout. |
| Randomization | Randomization was not performed as the collection of cells (FACS/Dispensing) was done randomly while recording cell size (where possible, as described in the methods section). |
| Blinding | The researchers were not blinded to the experimental conditions as it was not practically feasible. |

# Reporting for specific materials, systems and methods

We require information from authors about some types of materials, experimental systems and methods used in many studies. Here, indicate whether each material, system or method listed is relevant to your study. If you are not sure if a list item applies to your research, read the appropriate section before selecting a response.

### Materials & experimental systems

| n/a | Involved in the study |
|---|---|
| ☒ | ☐ Antibodies |
| ☐ | ☒ Eukaryotic cell lines |
| ☒ | ☐ Palaeontology and archaeology |
| ☒ | ☐ Animals and other organisms |
| ☒ | ☐ Human research participants |
| ☒ | ☐ Clinical data |
| ☒ | ☐ Dual use research of concern |

### Methods

| n/a | Involved in the study |
|---|---|
| ☒ | ☐ ChIP-seq |
| ☒ | ☐ Flow cytometry |
| ☒ | ☐ MRI-based neuroimaging |

# Eukaryotic cell lines

Policy information about cell lines

| | |
|---|---|
| Cell line source(s) | HEK293FT cells were purchased from Thermo Fisher. |
| Authentication | Cell lines were not recently authenticated. |
| Mycoplasma contamination | Cell lines were confirmed free of mycoplasma contamination using a PCR based test (Eurofins Genomics). |
| Commonly misidentified lines (See ICLAC register) | No commonly misidentified cell lines were used. |

