## [Peer Review File · Nature Methods]

Peer Review Information

Manuscript Title: Molecular spikes: a gold standard for single-cell RNA counting

Corresponding author name(s): Rickard Sandberg

Reviewer Comments & Decisions:

Decision Letter, initial version:

Subject: Decision on Nature Methods submission NMETH-BC45850

Message:

18th Jun 2021

Dear Dr Sandberg,

Your Brief Communication entitled "Molecular spikes: a gold standard for single-cell RNA counting" has now been seen by 3 reviewers, whose comments are attached. While they find your work of potential interest, they have raised serious concerns which in our view are sufficiently important that they preclude publication of the work in Nature Methods, at least in its present form.

As you will see, the reviewers raise concerns about the limited utility of the proposed spike-ins and whether they will be useful for corrections to transcript quantification in the context of existing RNA standards.

Should further experimental data allow you to fully address these criticisms we would be willing to look at a revised manuscript (unless, of course, something similar has by then been accepted at Nature Methods or appeared elsewhere). This includes submission or publication of a portion of this work somewhere else. We hope you understand that until we have read the revised paper in its entirety we cannot promise that it will be sent back for peer-review.

If you are interested in revising this manuscript for submission to Nature Methods in the future, please contact me to discuss your appeal before making any revisions. Otherwise, we hope that you find the reviewers' comments helpful when preparing your paper for submission elsewhere.

Sincerely,
Lei

Lei Tang, Ph.D.
Senior Editor
Nature Methods

Reviewers' Comments:

Reviewer #1:

Remarks to the Author:

In the study “Molecular spikes: a gold standard for single-cell RNA counting”, Ziegenhain et. al., they develop RNA spike-in controls that incorporate unique molecular identifiers. This allows the assembly of a reference set of synthetic controls that are then used to evaluate single-cell sequencing methods.

The incorporation of unique molecular identifiers (UMIs) within RNA spike-in controls represents the incorporation of a new functional element within spike-in controls that have been widely developed in recent years. This is achieved by the incorporation of degenerate oligonucleotides during the preparation of vector templates prior to in vitro transcription. The authors then demonstrate the utility of the spUMIs to benchmarking different single-cell sequencing protocols, highlighting the presence of technical artifacts and biases that occur.

Overall, I found the technical quality of the study sounds, the study design mature and the manuscript well prepared, with clear text and figures. However, my largest concern was that the study is currently quite narrow in scope, and would benefit from additional experiments that demonstrate the utility of the spUMIs in routine experimental analysis.

Whilst the authors show how the spike-in controls can be used to identify artefacts between different protocols, they do not demonstrate how or why the controls could be routinely used to improve the analysis of accompanying endogenous gene expression. At the moment, this means that the RNA spike-in controls act more like a reference material, rather than a new method to improve single-cell analysis. Whilst this authors demonstrate how the spike-in could be useful as a reference for benchmarking protocols, reagents or software, they not make a convincing case as to why scientists should routinely use these spike-ins to improve the interpretation or provide insight into their experiment.

MAJOR COMMENTS.

1. In this study, the authors usefully demonstrate how RNA spike-in controls can identify technical artefacts in a range of single-cell experimental protocols. However, a major benefit of spike-ins is that they can be added to a sample and act as an internal controls, enabling routine use during experiments. Whilst the RNA spike-in controls have been shown useful as a reference material for benchmarking different technologies, the authors do not show why the spike-ins should be routinely used to improve the performance of single experiment.

For example, the authors could show how the RNA spike-in controls can be used to improve the analysis of endogenous gene expression. The authors show with RNA spike-in controls that PCR amplification steps in single-cell protocols result in over-counting of genes. Do the RNA spike-in controls allow a user to mitigate this overcounting? Can a user normalise their gene-counts based on the over-counting measured by the RNA spike-in controls, possibly by subsampling or correcting the counting of accompanying genes? Is it possible to use the spUMIs to improve the analysis of the single-cell libraries using spike-in RNA controls. Such a demonstration would show that the RNA spike-in controls comprise a new method, rather than a reference material.

Accordingly, I would suggest that the authors use the RNA spike-ins during a single-cell experiment of a known biological model. This would allow the authors to show how the insights gained from the RNA spike-ins can improve the interpretation of the endogenous gene expression and provide biological insights. By stepping a reader through an experiment, they can show the utility of routinely using RNA spike-in controls in their research.

2. Description of the RNA spike-in controls. Whilst the study describes the design and manufacture of the spUMIs well, there is little information on the RNA spike-ins. For example, how many are manufactured in the mixture? Are they alternatively spliced? Is their concentration staggered to form a quantitative ladder etc. This could be useful information in comparing to alternative spike-in controls methods.

3. I think the title and abstract could be improved to better indicate to the reader what the study shows. The use of the term 'molecular spikes' seems quite vague and can refer to RNA, DNA or protein spike-in controls. For example, 'unique RNA spike-in controls' is a more precise description of the method.

Similarly, it also seems premature to claiming unique RNA spike-in controls as a gold standard. To become a gold-standard also depends on other factors beyond the study, including distribution and adoption of the controls. For example, the RNA spike-in controls may not become gold-standard, despite their superior advantages. Therefore, I think the authors should try to improve the title and the abstract to more clearly describe what is achieved within the study.

4. The authors claim that the RNA spike-in control with UMIs are superior to previous spike-ins (ERCC, SIRVs and Sequins). However, they do not provide any experimental comparisons with these previous techniques. While I agree that these unique spike-in controls have additional functionality, without additional details on the mixture size and composition, and ideally and experimental head-to-head comparison, it is difficult to evaluate against these alternative approaches.

MINOR COMMENTS.

5. A schematic figure to demonstrate the different steps in the single-cell protocols would be helpful to orientate readers that are not familiar with these techniques. For example, the paragraphs that describe the artifacts resulting from template switching in Smart-Seq3 could be easily described with a schematic figure that also shows how the spike-in RNA controls identify this artifact.

6. In the study, it seems that the authors were required to subsample the spike-in RNA controls by ten-fold to have a similar expression as endogenous genes. Is this because the spike-in amount was too high? Is achieving a correct dilution challenging in these protocols.

7. The authors provide the code for the analysis of the spUMIs and state that the RNA spike-in controls will be made available for researchers. However, this is a key point if they are to be widely adopted, and this accessibility should be highlighted within the abstract.

8. Whilst the authors usefully describe the use of the spike-in RNA controls for single-cell sequencing, the controls could be similarly useful for other applications. Could the authors also indicate how the spike-in controls could be used in conventional RNAseq and differential gene expression studies?

9. Line 30. The authors note that there is minimal overlap with mouse and human genomes. Could the authors please clarify what how this is defined?

Line 39. This paragraph is very long, and may benefit from being broken into two?

Line 43. The authors note that, on average, each UMI (which is 18nt long) has 1-2 errors. This sequencing error rate seems very high? Is this expected? And if so why/

Line 78. The authors state that “the RNA counting in SCR-seq was accurate – could the authors more clearly define and quantify this statement.

Line 144. follow > follows

Line 146. show > shows

Reviewer #2:

Remarks to the Author:

Ziegenhain and colleagues describe RNA spike-ins with built-in UMIs as means of benchmarking different single-cell RNA-seq protocols and measurements. This effectively offers synthetic transcripts with dual UMIs, which can be used to: i) diagnose library preparation problems such as priming of excess UMI-containing primers during PCR amplification steps, which I don't think would be possible to correct for computationally, and ii) benchmark computational corrections to molecular count estimates. The internal UMIs (spUMIs) are 18nt long, making it easy to correct for amplification/sequencing errors within them. The application is demonstrated on both popular full-length and 3' protocols. That said, the practical utility of the spike ins appears to be largely limited to the two aspects listed above.

The method is well-described. The analyses take a straightforward approach, providing clear results.

Major comments:

I only have one major comment, asking the authors to extend the discussion to note the limitations of the proposed spike-ins. Or, to be more precise to emphasize what lies outside of the intended scope. As it is mentioned in the text, early protocols have made extensive use of ERCC spike ins for a variety of purposes, including estimation of detection rates, variance normalization, etc. The related questions took a long time to settle. The spike ins described in the current manuscript are aimed at very specific applications and cannot be used to address a variety of related effects. It would be very useful for the readers to come out with a clear understanding of that. For instance,

- The proposed spike-ins cannot be used to diagnose gene-specific detection biases. Even a perfect agreement of spUMIs and UMI counts, like the one shown in Fig. 1d for Smart-seq3 does not mean that the detection rates of the spiked in molecules was linear with respect to their true concentration.
- The proposed spike-ins do not provide obvious means to normalize expression variability between cells/droplets.

Similarly, it would be great to mention that the proposed spike-ins do not provide obvious help in troubleshooting issues with protocol steps prior to UMI incorporation. This is not a criticism of the approach, but simply a request for a clarification.

Minor comments

1. Fig. 1h is somewhat problematic and, at least to me, not very useful. Graphically, the shaded regions are not visible, and the no-exonuclease/SCRB-seq lines are hardly distinguishable. It's unclear what the

dotted line means. Overall, the point that there's higher error rates for highly expressed molecules is already obvious from Fig. 1g, so I am not sure 1h adds much.

2. The setup in Supp. Fig. 5 is excellent, but the plots are difficult to interpret. For the left plot, the legend says that "mean counting difference" is shown, but given that the value shown is on the log₁₀ scale, what's shown on the left plot is not clear (e.g. what are negative values? is there a pseudocount? where is a region of UMI undercounting?). Perhaps it's best to write out a formula for the values being shown. The right plots seem to be mostly showing rendering artifacts – perhaps contour plot is not the best choice.

3. line 107: "we observe that UMIs of a length 6 or lower reach significant collision rates leading to under-counting even in the absence of UMI error correction (Figure 2a, b)". I am guessing the authors imply that UMI correction methods would make the counts even smaller. This is a bit confusing, however, as not all of the correction methods do that. Some, like ref.9 try to adjust for collision rates. It would be interesting, by the way, to see if it's actually able to account for underestimation in a benchmark like Fig 2c,d.

4. line 52: "we estimated the complexity of the total 5'-molecular spike pool to 3.2 million, demonstrating that there were no unexpected bottlenecks in the cloning and production procedure (Figure 1c)." It's not really clear where the expectation about the total complexity comes from. If the authors simply want to point out that there was a good yield, perhaps the sentence should be rephrased.

5. line 147: "The literature provides conflicting recommendations regarding UMI lengths^{12,17}, as longer UMIs can interfere with method sensitivity and shorter UMIs have limited coding capacity." Perhaps I've missed something in these papers, but in general, I thought the argument was long settled – longer UMIs provide better ways of correcting for counting problems. This is in fact illustrated by the ease with which the authors were argue for "ground truth" using long 18nt spUMIs.

Reviewer #3:

Remarks to the Author:

Summary:

Ziegnhain, Hendriks, et al develop "molecular spikes", synthetic RNA spike-ins with built-in UMIs, to evaluate the accuracy of transcript quantification by different single-cell sequencing (scRNA-seq) protocols.

The authors apply "molecular spikes" in order to identify potential transcript quantification inaccuracies in different scRNA-seq library protocols including Smart-seq3, 10X 3'-sequencing, and plate-based 3'-sequencing (SCRB-seq). Encouragingly, the authors find that the inferred molecular counts are generally in good agreement with their "molecular spikes". The authors also applied "molecular spikes" to a direct PCR condition for a more recently developed tSCRB-seq protocol to find that this particular protocol resulted in substantial overcounting.

The finding that transcript quantifications from more commonly used scRNA-seq protocols such as 10X 3'-sequencing and Smart-Seq3 are generally accurate (high correlation and low inflation with respect to "molecular spikes") is very reassuring and will be of high interest to the single-cell community. However, the utility of "molecular spikes" as a tool seems more restricted to the development of new scRNA-seq protocols. Ultimately, how "molecular spikes" may enable corrections to transcript quantification in scRNA-seq data and how such corrections may enable biological discoveries is currently not well delineated.

Major comments:

1. It is unclear how "molecular spikes" as a tool for evaluating the accuracy of transcript quantification by scRNA-seq protocols differs from previously published and available RNA spike-in tools such as ERCC, etc. Is there a reason to prefer "molecular spikes" over these other spike-ins?
2. The ground truth transcript abundance range for the spiked molecules in "molecular spikes" all seem quite high e.g. > 50 counts (according to figure 1). Many in the field are interested in quantifying the transcript abundance of lowly expressed genes such as transcription factors. Can "molecular spikes" evaluate the accuracy of transcript quantification by different scRNA-seq protocols for such lowly expressed genes (ex. 1 to 10, 10 to 50 counts range)?
3. The authors note that "the reported increased sensitivity obtained in tSCRB-seq is completely artificial due to the removal of the essential cDNA cleanup step." (Line 92). Is there a quantification to show that after correction with "molecular spikes", the sensitivity of tSCRB-seq is comparable to the old SCRB-seq and thus the previously reported increased sensitivity is "completely artificial"? Generally, given the strength of this statement, further quantitative substantiation is warranted.
4. If overcounting is identified using "molecular spikes" such as in the tSCRB-seq benchmark, how do the authors suggest correcting for such overcounting? Does the correction need to be applied to the protocol stage or can be correction by achieved computationally? Does such overcounting always linearly follows sequencing depth such that the overcounted amount could be subtracted or regressed out?

Minor comments:

1. Does correction using "molecular spikes" enable the discovery of new cell-types or cell-states that are not apparent without correction in the tSCR-seq data?
2. Do "molecular spikes" provide a better correction of transcript quantification than previously proposed computational correction strategies?

Decision Letter, initial version – appeal:

Subject: Decision on appeal of Nature Methods submission NMETH-BC45850A-Z

Message:

5th Jul 2021

Dear Dr Sandberg,

Thank you for your letter asking us to reconsider our decision on your Brief Communication, "Molecular spikes: a gold standard for single-cell RNA counting". After careful consideration we have decided that we are willing to consider a revised version of your manuscript that includes additional experiments to strengthen the utility of the molecular spikes.

- * include a point-by-point response to our referees and to any editorial suggestions
- * please underline/highlight any additions to the text or areas with other significant changes to facilitate review of the revised manuscript
- * address the points listed described below to conform to our open science requirements
- * ensure it complies with our general format requirements as set out in our guide to authors at www.nature.com/naturemethods
- * resubmit all the necessary files electronically by using the link below to access your home page

[REDACTED]

We hope to receive your revised paper within four months. If you cannot send it within this time, please let us know. In this event, we will still be happy to reconsider your paper at a later date so long as nothing similar has been accepted for publication at Nature Methods or published elsewhere.

OPEN SCIENCE REQUIREMENTS

REPORTING SUMMARY AND EDITORIAL POLICY CHECKLISTS

When revising your manuscript, please submit reporting summary and editorial policy checklists.

IMAGE INTEGRITY

- that unprocessed scans are clearly labelled and match the gels and western blots presented in figures.
- that control panels for gels and western blots are appropriately described as loading on sample processing controls

-- all images in the paper are checked for duplication of panels and for splicing of gel lanes.

DATA AVAILABILITY

Please include a “Data availability” subsection in the Online Methods. This section should inform readers about the availability of the data used to support the conclusions of your study, including accession codes to public repositories, references to source data that may be published alongside the paper, unique identifiers such as URLs to data repository entries, or data set DOIs, and any other statement about data availability. At a minimum, you should include the following statement: “The data that support the findings of this study are available from the corresponding author upon request”, describing which data is available upon request and mentioning any restrictions on availability. If DOIs are provided, please include these in the Reference list (authors, title, publisher (repository name), identifier, year). For more guidance on how to write this section please see: <http://www.nature.com/authors/policies/data/data-availability-statements-data-citations.pdf>

CODE AVAILABILITY

Please include a “Code Availability” subsection in the Online Methods which details how your custom code is made available. Only in rare cases (where code is not central to the main conclusions of the paper) is the statement “available upon request” allowed (and reasons should be specified).

Authors reporting mutant strains and cell lines are strongly encouraged to use established public repositories.

SUPPLEMENTARY PROTOCOL

To help facilitate reproducibility and uptake of your method, we ask you to prepare a step-by-step Supplementary Protocol for the method described in this paper. We [encourage authors to share their step-by-step experimental protocols](https://www.nature.com/nature-research/editorial-policies/reporting-standards#protocols) on a protocol sharing platform of their choice and report the protocol DOI in the reference list. Nature Research's Protocol Exchange is a free-to-use and open resource for protocols; protocols deposited in Protocol Exchange are citable and can be linked from the published article. More details can found at www.nature.com/protocolexchange/about.

ORCID

Best regards,
Lei

Lei Tang, Ph.D.
Senior Editor
Nature Methods

Author Rebuttal to Initial comments

Reviewers' Comments:**Reviewer #1:**

In the study "Molecular spikes: a gold standard for single-cell RNA counting", Ziegenhain et. al., they develop RNA spike-in controls that incorporate unique molecular identifiers. This allows the assembly of a reference set of synthetic controls that are then used to evaluate single-cell sequencing methods.

The incorporation of unique molecular identifiers (UMIs) within RNA spike-in controls represents the incorporation of a new functional element within spike-in controls that have been widely developed in recent years. This is achieved by the incorporation of degenerate oligonucleotides during the preparation of vector templates prior to in vitro transcription. The authors then demonstrate the utility of the spUMIs to benchmarking different single-cell sequencing protocols, highlighting the presence of technical artifacts and biases that occur.

Overall, I found the technical quality of the study sounds, the study design mature and the manuscript well prepared, with clear text and figures. However, my largest concern was that the study is currently quite narrow in scope, and would benefit from additional experiments that demonstrate the utility of the spUMIs in routine experimental analysis.

Whilst the authors show how the spike-in controls can be used to identify artefacts between different protocols, they do not demonstrate how or why the controls could be routinely used to improve the analysis of accompanying endogenous gene expression. At the moment, this means that the RNA spike-in controls act more like a reference material, rather than a new method to improve single-cell analysis. Whilst this authors demonstrate how the spike-in could be useful as a reference for benchmarking protocols, reagents or software, they not make a convincing case as to why scientists should routinely use these spike-ins to improve the interpretation or provide insight into their experiment.

We thank the Reviewer for their appreciation of our work, including its conceptual novelty, and for their constructive and insightful feedback. Inspired by the Reviewer's comments, we designed a complex set of 264 molecular spikes (with inbuilt spUMIs) to demonstrate how they improve routine experimental analysis. As seen in the detailed responses below, we show that the set of molecular spikes enable the computationally rescue of faulty experiments with severe counting problems (or no UMI-based counting used at all). Moreover, the widespread inclusion of molecular spikes would enable the inference of the endogenous mRNA amounts per cell important for estimating the variation in mRNA amounts over cell types and tissues. As seen in the new experiments, the detected UMI counts per cell were only modestly correlating with cell diameters, but scaling the counts using the molecular spikes revealed a large-scale effect on endogenous mRNA amounts. The new set of molecular spikes and the new experiments demonstrate general benefits that would be valuable in all scRNA-seq experiments, in particular large-scale projects (e.g. BICCN or Human Cell Atlas) that aims to identify the molecular underpinning of cell type variation with tissues in mammals.

MAJOR COMMENTS.

1. In this study, the authors usefully demonstrate how RNA spike-in controls can identify technical artefacts in a range of single-cell experimental protocols. However, a major benefit of spike-ins is that they can be added to a sample and act as an internal controls, enabling routine use during experiments. Whilst the RNA spike-in controls have been shown useful as a referenc material for benchmarking different technologies, the authors do not show why the spike-ins should be routinely used to improve the performance of single experiment.

For example, the authors could show how the RNA spike-in controls can be used to improve the analysis of endogenous gene expression. The authors show with RNA spike-in controls that PCR amplification steps in single-cell protocols result in over-counting of genes. Do the RNA spike-in controls allow a user to mitigate this overcounting? Can a user normalise their gene-counts based on the over-counting measured by the RNA spike-in controls, possibly by subsampling or correcting the counting of accompanying genes? Is it possible to use the spUMIs to improve the analysis of the single-cell libraries using spike-in RNA controls. Such a demonstration would show that the RNA spike-in controls comprise a new method, rather than a reference material.

Accordingly, I would suggest that the authors use the RNA spike-ins during a single-cell experiment of a known biological model. This would allow the authors to show how the insights gained from the RNA spike-ins can improve the interpretation of the endogenous gene expression and provide biological insights. By stepping a reader through an experiment, they can show the utility of routinely using RNA spike-in controls in their research.

We thank the reviewer for their detailed comments and their appreciation for the novel possibilities to determine counting accuracy using molecular spikes. To address the remaining concerns, we have extensively revised our spike-in design and created a new set of molecular spikes comprised of 11 distinct transcript sequences that span different lengths (500-3000 bp) and GC contents (40-60 %). Within each of these 11 different spike-in sequences, we included both the spUMI for ground-truth counting and also 24 specific 7 nt barcodes that encode for varying expression levels over 12 steps of 2-fold difference. Thus, within each distinct spike-in sequence we can create an expression standard curve. Elegantly, the expression levels of every barcode were accurately derived from counting using the spUMI, and were not relying on error-prone measurements or cloning.

Reviewers Figure 1 (Figure 3a in the revised manuscript). (a) Illustration of the design of the complex set of molecular spike-ins. The set consists of 264 unique spike-ins based on 11 distinct spike sequences of different lengths and GC levels, as shown. Each of these 11 sequences further contains 24 different barcodes that were introduced in titration to obtain a standard curve per spike in sequence (using 2 barcodes per expression level, covering 12 different expression levels).

We show that the resulting set of 264 spike-ins (24 barcodes x 11 sequences) span the observed expression levels in human cells well and model technical variability excellently (new Figure 3c, pasted below). These features and the fact that the exact molecular abundances of the spikes can be counted using the spUMIs makes them ideal as a replacement for the most commonly used ERCC spikes and for spike-in based normalization & biological variability classification.

Reviewers Figure 2 (Figure 3c in revised manuscript): (c) Scatter plot showing the mean molecules detected per cell (x-axis) against the squared CV (y-axis).

Importantly, we next showed that faulty RNA counting in scRNA-seq experiments can be computationally rescued using the new set of molecular spikes. We prepared scRNA-seq data from HEK293FT cells using Smart-seq3xpress, and we used the raw read counts per gene as an example of heavily inflated counts. Note the read counts had not undergone any kind of UMI-based correction. Instead, the UMI-based counting was used as the “ground truth” RNA counts that the rescue was compared against. Using the molecular spikes, to capture the relationship between spike-in and endogenous mRNA amounts in cells, we modified a recently described quantile-normalization method (Townes and Irizarry 2020) to infer quasi-UMIs. Strikingly, the correction strategy we developed inferred quasi-UMIs that closely resembled the conventional UMI counts (Figure 3h,i,j, also pasted below). Indeed, this strategy should enable quasi-UMIs RNA counts even for experiments that completely lacked UMIs as in widely used Smart-seq2. We believe this experiment provides a strong argument of the routine use of the new molecular spikes.

Reviewers Figure 3 (Figure 3h,i,j in revised manuscript): (h) Distributions of non-zero raw reads, inferred quasi-UMIs and observed Smart-seq3 UMIs across genes, shown for 20 representative HEK cells. (i) Boxplots showing the total number of reads, inferred quasi-UMIs and observed Smart-seq3 UMIs per cell ($n = 151$). (j) Scatter plot of observed Smart-seq3 UMIs against inferred quasi-UMIs across all genes ($n = 17,054$) for a representative cell.

Finally, we also demonstrate how transcriptome alterations associated with cellular properties could be better captured by using the molecular spikes. While sorting HEK293FT cells of smaller and larger cell diameters into individual wells for cell lysis, we recorded cell diameters of each cell using an F.SIGHT. The overall relationship between UMI-based RNA counts and cell diameter was modest (new

Figure 3d, also pasted below), since UMI-based RNA counts were heavily associated with cellular sequence depths (new Supplementary Figure 7e). However, after molecular spike-based normalizations, large differences in transcriptomes between small and large HEK cells became apparent (Figure 3e,f,g, also pasted below). The ability to infer total transcriptome sizes per cell using the molecular spikes additionally argues for their widespread use, since revealing how transcriptome sizes vary with cell types provides important molecular insights into cellular variation in tissues.

Reviewers Figure 3 (Figure 3d,e,f,g in revised manuscript): (d) Scatter plot showing the observed number of RNA counts of cellular genes (UMIs) against cell diameter, which was recorded while dispensing each cell into wells. Linear regression shown with line. (e) Scatter plot of percent reads aligning to spike-ins per cell against the recorded cell diameter. Linear regression shown as line. (f) Scatter plot showing inferred cellular RNA counts against recorded cell diameter, with linear regression shown as line. (g) Boxplots showing observed (left) and inferred (right) cellular RNAs for large (>25um) and small (<25um) HEK cells (n = 84 and 67, respectively).

2. Description of the RNA spike-in controls. Whilst the study describes the design and manufacture of the spUMIs well, there is little information on the RNA spike-ins. For example, how many are manufactured in the mixture? Are they alternatively spliced? Is their concentration staggered to form a quantitative ladder etc. This could be useful information in comparing to alternative spike-in controls methods.

We have added more information on the design throughout the text to clarify the specific information per RNA spike-ins. Most importantly, we included a new figure that illustrate the design of the new set of molecular spikes (which is Figure 3a in the revised manuscript, also pasted below). Additionally, we now provide more details in the relevant locations of the revised manuscript. The quantitative ladder design, within each spike-in sequence, is presented in Supplementary Figure 9 and more information per spike-in sequence was summarized in Supplementary Table 1.

Reviewers Figure 4 (Figure 3a in the revised manuscript). (a) Illustration of the design of the complex set of molecular spike-ins. The set consists of 264 unique spike-ins based on 11 distinct spike sequences of different lengths and GC levels, as shown. Each of these 11 sequences further contains 24 different barcodes that were introduced in titration to obtain a standard curve per spike in sequence (using 2 barcodes per expression level, covering 12 different expression levels).

Reviewers Figure 5 (Supplementary Figure 9 of the revised manuscript): Scatter plots showing the obtained spUMI counts per 11 spike-in designs, and across the 24 barcoded transcripts per spike-in sequence design.

3. I think the title and abstract could be improved to better indicate to the reader what the study shows. The use of the term 'molecular spikes' seems quite vague and can refer to RNA, DNA or protein spike-in controls. For example, 'unique RNA spike-in controls' is a more precise description of the method.

Similarly, it also seems premature to claiming unique RNA spike-in controls as a gold standard. To become a gold-standard also depends on other factors beyond the study, including distribution and adoption of the controls. For example, the RNA spike-in controls may not become gold-standard, despite their superior advantages. Therefore, I think the authors should try to improve the title and the abstract to more clearly describe what is achieved within the study.

We respectfully disagree with this comment. Since the title contains the reference to single-cell RNA counting, the term “molecular spikes” can realistically only refer to RNA controls. While we are of course eager for the research community to adopt the molecular spikes in routine scRNA-seq experiments, the level of uptake of the spike-ins is to be seen. What is clear however, is that these molecular spikes overcome a 10-year-old challenge in the field to actually experimentally control the fidelity of the ubiquitously used UMI based RNA counting schemes. Therefore it is clear that the molecular spikes are the new gold standard.

4. The authors claim that the RNA spike-in control with UMIs are superior to previous spike-ins (ERCC, SIRVs and Sequins). However, they do not provide any experimental comparisons with these previous techniques. While I agree that these unique spike-in controls have additional functionality, without additional details on the mixture size and composition, and ideally and experimental head-to-head comparison, it is difficult to evaluate against these alternative approaches.

We designed the molecular spikes for unique capabilities to validate RNA counting schemes in single-cell applications. The ability to experimentally determine their exact molecular abundances, by simply sequencing their barcodes and inbuilt spUMIs makes the quantitative information present for these spikes much more precise than previous spike-in designs mentioned by the reviewer. In the light of this comment, we have rephrased manuscript text to make sure the reader does not get the impression that molecular spikes are always superior to existing spike-ins. Rather, we highlight the unique and highly important feature of assessing molecule counting accuracy across single-cell RNA-seq methods, the ability to correct inflated counts to quasi-UMIs, and more accurately infer endogenous mRNA amounts.

MINOR COMMENTS.

5. A schematic figure to demonstrate the different steps in the single-cell protocols would be helpful to orientate readers that are not familiar with these techniques. For example, the paragraphs that describe the artifacts resulting from template switching in Smart-Seq3 could be easily described with a schematic figure that also shows how the spike-in RNA controls identify this artifact.

We thank the reviewer for the suggestion. A schematic of the protocol workflows has been added to the revised manuscript (new Supplementary Figure 2d,e).

Reviewers Figure 6. (Supplementary Figures 2d,e in the revised manuscript). (d) Schematic overview of the counting-critical steps in the Smart-seq3 protocol. **(e)** Schematic overview of the evaluated protocol variations for the SCR-seq library preparation.

6. In the study, it seems that the authors were required to subsample the spike-in RNA controls by ten-fold to have a similar expression as endogenous genes. Is this because the spike-in amount was too high? Is achieving a correct dilution challenging in these protocols.

It is correct that the initial design of the molecular spikes was intentionally added at high copy numbers, and subsampling was employed to check a wide range of expression levels. The significantly improved design and generation of the new complex set of molecular spikes has a dilution series within each of the 11 spike-in sequences. In the experiments shown in Figure 3, the abundances of the 264 spike-in sequences span well the physiologically relevant expression levels of endogenous genes (see Figure 3c). Typically, correctly diluting and pipetting spike-ins to each cell is critical for their use, however with molecular spikes the ability to directly count the spiked-in RNAs in each cell (using their spUMIs) significantly lowers these problems for users. This is another great advantage of using these self-controlling molecular spikes.

7. The authors provide the code for the analysis of the spUMIs and state that the RNA spike-in controls will be made available for researchers. However, this is a key point if they are to be widely adopted, and this accessibility should be highlighted within the abstract.

We thank the reviewer for appreciating the code availability. We have mentioned sharing of the spikes and code in the revised abstract.

8. Whilst the authors usefully describe the use of the spike-in RNA controls for single-cell sequencing, the controls could be similarly useful for other applications. Could the authors also indicate how the spike-in controls could be used in conventional RNAseq and differential gene expression studies?

Since the main application of UMIs in terms of RNA sequencing is within the area of single-cell sequencing, we have not applied molecular spikes in the context of bulk RNA-seq. Given that bulk RNA-

seq protocols without UMIs are well established and widely used, we do not see an urgent need for the application of molecular spikes.

9. Line 30. The authors note that there is minimal overlap with mouse and human genomes. Could the authors please clarify what how this is defined?

In terms of the initial spike-in design, candidate sequences were compared to the mouse and human genomes using BLAST to exclude matches. In the design of the new complex molecular spikes set, we first mapped simulated sequencing reads for candidate sequences against the human genome with STAR and then confirmed absence of matches in mouse and human with BLAST. We have added these details to the methods section of the revised manuscript.

Line 39. This paragraph is very long, and may benefit from being broken into two?

We have revised the text to ensure it is easy to follow throughout.

Line 43. The authors note that, on average, each UMI (which is 18nt long) has 1-2 errors. This sequencing error rate seems very high? Is this expected? And if so why/

This observation stems from the fact that due to deep sequencing, most of the UMIs of 18 nt length are observed with many read counts. Thus, at the usual 0.5-1% error rate in NGS experiments (RT errors + PCR errors + sequencing errors), we do expect one or two mismatches in at least one of those reads, necessitating error correction at hamming distance 1 or 2.

Line 78. The authors state that “the RNA counting in SCRB-seq was accurate – could the authors more clearly define and quantify this statement.

We have more precisely quantitated this statement in the main text:
“The RNA counting in SCRB-seq was accurate with on average only 0.1 % of deviance to spUMI ground truth (Figure 1g).”

Line 144. follow > follows
We fixed the typo.

Line 146. show > shows
We fixed the typo.

Reviewer #2:

Remarks to the Author:

Ziegenhain and colleagues describe RNA spike-ins with built-in UMIs as means of benchmarking different single-cell RNA-seq protocols and measurements. This effectively offers synthetic transcripts with dual UMIs, which can be used to: i) diagnose library preparation problems such as priming of excess UMI-containing primers during PCR amplification steps, which I don't think would be possible to correct for computationally, and ii) benchmark computational corrections to molecular count estimates. The internal UMIs (spUMIs) are 18nt long, making it easy to correct for amplification/sequencing errors within them. The application is demonstrated on both popular full-length and 3' protocols. That said, the practical utility of the spike ins appears to be largely limited to the two aspects listed above.

The method is well-described. The analyses take a straightforward approach, providing clear results.

Major comments:

I only have one major comment, asking the authors to extend the discussion to note the limitations of the proposed spike-ins. Or, to be more precise to emphasize what lies outside of the intended scope. As it is mentioned in the text, early protocols have made extensive use of ERCC spike ins for a variety of purposes, including estimation of detection rates, variance normalization, etc. The related questions took a long time to settle. The spike ins described in the current manuscript are aimed at very specific applications and cannot be used to address a variety of related effects. It would be very useful for the readers to come out with a clear understanding of that. For instance,

- The proposed spike-ins cannot be used to diagnose gene-specific detection biases. Even a perfect agreement of spUMIs and UMI counts, like the one shown in Fig. 1d for Smart-seq3 does not mean that the detection rates of the spiked in molecules was linear with respect to their true concentration.
- The proposed spike-ins do not provide obvious means to normalize expression variability between cells/droplets.

We thank the reviewer for their comments and their appreciation for the novel possibilities to determine counting accuracy using molecular spikes. To address the remaining concerns, we have extensively revised our spike-in design and created a new set of molecular spikes comprised of 11 distinct transcript sequences that span different lengths (500-3000 bp) and GC contents (40-60 %). Within each of these 11 different spike-in sequences, we included both the spUMI for ground-truth counting and also 24 specific 7 nt barcodes that encode for varying expression levels over 12 steps of 2-fold difference. Thus, within each distinct spike-in sequence we can create an expression standard curve. Elegantly, the expression levels of every barcode were accurately derived from counting using the spUMI, and were not relying on error-prone measurements or cloning.

Reviewers Figure 1 (Figure 3a in the revised manuscript). (a) Illustration of the design of the complex set of molecular spike-ins. The set consists of 264 unique spike-ins based on 11 distinct spike sequences of different lengths and GC levels, as shown. Each of these 11 sequences further contains 24 different

barcodes that were introduced in titration to obtain a standard curve per spike in sequence (using 2 barcodes per expression level, covering 12 different expression levels).

We show that the resulting set of 264 spike-ins (24 barcodes x 11 sequences) span the observed expression levels in human cells well and model technical variability excellently (new Figure 3c, pasted below). These features and the fact that the exact molecular abundances of the spikes can be counted using the spUMIs makes them ideal as a replacement for the most commonly used ERCC spikes and for spike-in based normalization & biological variability classification.

Reviewers Figure 2 (Figure 3c in revised manuscript): (c) Scatter plot showing the mean molecules detected per cell (x-axis) against the squared CV (y-axis).

Importantly, we next showed that faulty RNA counting in scRNA-seq experiments can be computationally rescued using the new set of molecular spikes. We prepared scRNA-seq data from HEK293FT cells using Smart-seq3^{xpress}, and we used the raw read counts per gene as an example of heavily inflated counts. Note the read counts had not undergone any kind of UMI-based correction. Instead, the UMI-based counting was used as the “ground truth” RNA counts that the rescue was compared against. Using the molecular spikes, to capture the relationship between spike-in and endogenous mRNA amounts in cells, we modified a recently described quantile-normalization method (Townes and Irizarry 2020) to infer quasi-UMIs. Strikingly, the correction strategy we developed based on the molecular spike to inferred quasi-UMIs closely resembling conventional UMI counts (Figure 3h,i,j, also pasted below). Indeed, this strategy should enable quasi RNA counts even for experiments that completely lacked UMIs as in widely used Smart-seq2. We believe this experiment provides a strong argument of the routine use of the new molecular spikes.

Reviewers Figure 3 (Figure 3h,i,j in revised manuscript): (h) Distributions of non-zero raw reads, inferred quasi-UMIs and observed Smart-seq3 UMIs across genes, shown for 20 representative HEK cells. (i) Boxplots showing the total number of reads, inferred quasi-UMIs and observed Smart-seq3 UMIs per cell (n = 151). (j) Scatter plot of observed Smart-seq3 UMIs against inferred quasi-UMIs across all genes (n = 17,054) for a representative cell.

Finally, we also demonstrate how transcriptome alterations associated with cellular properties could be better captured by using the molecular spikes. While sorting HEK293FT cells or smaller and larger cell diameters into individual wells for cell lysis, we recorded cell diameters of each cell using an F.SIGHT. The overall relationship between UMI-based RNA counts and cell diameter was modest (new Figure 3d, also pasted below), since UMI-based RNA counts were heavily associated with cellular sequencing depths (new Supplementary Figure 7e). However, after molecular spike-based normalizations, large differences in transcriptomes between small and large HEK cells became apparent (Figure 3e,f,g, also pasted below). The ability to infer total transcriptome sizes per cell using the molecular spikes additionally argues for their widespread use, since revealing how transcriptome sizes vary with cell types provides important molecular insights into cellular variation in tissues.

Reviewers Figure 4 (Figure 3d,e,f,g in revised manuscript): (d) Scatter plot showing the observed number of RNA counts of cellular genes (UMIs) against cell diameter, which was recorded while dispensing each cell into wells. Linear regression shown with line. (e) Scatter plot of percent reads aligning to spike-ins per cell against the recorded cell diameter. Linear regression shown as line. (f) Scatter plot showing inferred cellular RNA counts against recorded cell diameter, with linear regression shown as line. (g) Boxplots showing observed (left) and inferred (right) cellular RNAs for large (>25um) and small (<25um) HEK cells (n = 84 and 67, respectively).

Similarly, it would be great to mention that the proposed spike-ins do not provide obvious help in troubleshooting issues with protocol steps prior to UMI incorporation. This is not a criticism of the approach, but simply a request for a clarification.

We added a sentence to the discussion to clarify this exact point.

Minor comments

1. Fig. 1h is somewhat problematic and, at least to me, not very useful. Graphically, the shaded regions are not visible, and the no-exonuclease/SCRB-seq lines are hardly distinguishable. It's unclear what the dotted line means. Overall, the point that there's higher error rates for highly expressed molecules is already obvious from Fig. 1g, so I am not sure 1h adds much.

The reviewer correctly points out that Figure 1g already show that more errors in counting is found for more highly expressed genes. We do think that Figure 1h is important, since it explains the exact nature of the gross miscounting in tSCRB-seq, since we observed a linear relationship between sequence depth per molecule and the level of overcounting, independent on the expression level of molecules. We found that the new counts follow closely the number of sequenced reads (that linear relationship being drawn as the dotted line).

We thank the reviewer for pointing out the lack of clarity surrounding this important result. To improve clarity, we increased the width of the SCR-seq/No exonuclease lines. Furthermore, we have clarified the result text and added a description of the dotted line to the Figure legend.

Revised result text:

"In fact, the "direct PCR" implementation in tSCRB-seq introduced new UMIs nearly in every new sequenced read, resulting in overcounting that linearly follows sequencing depth irrespective of expression level (Figure 1h)."

New sentence in Figure legend:

"The dotted line represents the expected overcounting if every sequenced read corresponds to a new UMI observation."

2. The setup in Supp. Fig. 5 is excellent, but the plots are difficult to interpret. For the left plot, the legend says that "mean counting difference" is shown, but given that the value shown is on the log10 scale, what's shown on the left plot is not clear (e.g. what are negative values? is there a pseudocount? where is a region of UMI undercounting?). Perhaps it's best to write out a formula for the values being shown. The right plots seem to be mostly showing rendering artifacts – perhaps contour plot is not the best choice.

We agree with the reviewer and have further improved the figure rendering and description. The contour plots indeed showed rendering artifacts in certain PDF viewers. We have made the plots more visible by rasterizing them so that the viewing should be reproducible.

In the left panels, the absolute count deviation from ground truth was colored after log10 transformation with a pseudocount of 1. Hence, small negative values simply denote low average undercounting from ground truth which, given these results were generated from the 10x Genomics dataset with long UMIs is rarely expected.

To clarify the scale of the contour plots, we have amended the figure legend:

“In each of the contour plots, the left panel is colored by the deviation from ground truth counting on a log10 scale with a pseudocount of 1 added and the right side denotes the deviation from ground truth relative to the mean expression level.”

3. line 107: “we observe that UMIs of a length 6 or lower reach significant collision rates leading to under-counting even in the absence of UMI error correction (Figure 2a, b)”. I am guessing the authors imply that UMI correction methods would make the counts even smaller. This is a bit confusing, however, as not all of the correction methods do that. Some, like ref.9 try to adjust for collision rates. It would be interesting, by the way, to see if it's actually able to account for underestimation in a benchmark like Fig 2c,d.

We have clarified the wording in the main text:

“In contrast to a previous report¹², we observe that UMIs of a length 6 or lower had significant collision rates leading to under-counting even before applying UMI error corrections which would lead to a further reduction in counts (Figure 2a, b). While empirical Bayesian correction algorithms have proposed to account for the lower coding capacity, their run time is prohibitive for larger datasets.”

4. line 52: “we estimated the complexity of the total 5'-molecular spike pool to 3.2 million, demonstrating that there were no unexpected bottlenecks in the cloning and production procedure (Figure 1c).” It's not really clear where the expectation about the total complexity comes from. If the authors simply want to point out that there was a good yield, perhaps the sentence should be rephrased.

Point taken. We rephrased the sentence to make it more clear that it is not just about yield, but about performing the cloning of the spUMI segment into the spike-in sequence in a way that reduces the bottleneck for the number of observable molecules. We added the following sentence to the result section:

“Because the number of bacterial clones transformed is the main determinant of library complexity, we estimated the complexity of the total 5'-molecular spikes to 3.2 million by fitting an asymptotic non-linear model to the number of observed spUMIs sequences across cells (Figure 1c).”

5. line 147: “The literature provides conflicting recommendations regarding UMI lengths^{12,17}, as longer UMIs can interfere with method sensitivity and shorter UMIs have limited coding capacity.” Perhaps I've missed something in these papers, but in general, I thought the argument was long settled – longer UMIs provide better ways of correcting for counting problems. This is in fact illustrated by the ease with which the authors were argue for “ground truth” using long 18nt spUMIs.

We have clarified these statements in the revised text to clarify that while the expectation on longer UMIs providing more coding capacity is straightforward, the implementation of arbitrary lengths may not always be feasible in the context of scRNA-seq:

“The literature provides conflicting recommendations regarding UMI lengths, and finding the optimal compromise for scRNA-seq applications is not straightforward as longer UMIs can interfere with method sensitivity by decreasing the efficiency of reactions (eg. reverse transcription) and shorter UMIs have limited coding capacity.”

Reviewer #3:

Remarks to the Author:

Summary:

Ziegnhain, Hendriks, et al develop "molecular spikes", synthetic RNA spike-ins with built-in UMIs, to evaluate the accuracy of transcript quantification by different single-cell sequencing (scRNA-seq) protocols.

The authors apply "molecular spikes" in order to identify potential transcript quantification inaccuracies in different scRNA-seq library protocols including Smart-seq3, 10X 3'-sequencing, and plate-based 3'-sequencing (SCRIB-seq). Encouragingly, the authors find that the inferred molecular counts are generally in good agreement with their "molecular spikes". The authors also applied "molecular spikes" to a direct PCR condition for a more recently developed tSCRIB-seq protocol to find that this particular protocol resulted in substantial overcounting.

The finding that transcript quantifications from more commonly used scRNA-seq protocols such as 10X 3'-sequencing and Smart-Seq3 are generally accurate (high correlation and low inflation with respect to "molecular spikes") is very reassuring and will be of high interest to the single-cell community. However, the utility of "molecular spikes" as a tool seems more restricted to the development of new scRNA-seq protocols. Ultimately, how "molecular spikes" may enable corrections to transcript quantification in scRNA-seq data and how such corrections may enable biological discoveries is currently not well delineated.

Major comments:

1. It is unclear how "molecular spikes" as a tool for evaluating the accuracy of transcript quantification by scRNA-seq protocols differs from previously published and available RNA spike-in tools such as ERCC, etc. Is there a reason to prefer "molecular spikes" over these other spike-ins?

The molecular spikes add for the first time the ability to directly measure the fidelity of molecule counting procedures which have become ubiquitous in scRNA-seq, uniquely possible due to the inbuilt UMI within the spikes themselves. As opposed to ERCCs which have so far relied on the correlating read counts to estimated molar concentrations, the molecular spikes allow for a direct comparison of the spUMI counting to the UMI of the library preparation method. This fundamental difference is necessary in order to make statements on the accuracy of the UMI counting capabilities of scRNA-seq methods. In contrast to the initial submission, we have now expanded the set of molecular spikes to provide a more diverse set of sequences spanning, GC, lengths and expression levels, thus making them a routinely applicable replacement for previous spike-ins like ERCC. Furthermore, the revised set of molecular spikes also comes with the possibility to compute quasi-UMIs in cases of overcounting, enabling researchers to "rescue" experiments with count inflation. We have added a better description of what the spike-ins comprise in the text along with more descriptive information (new Supplementary Table 1) to better guide readers through these improvements.

2. The ground truth transcript abundance range for the spiked molecules in "molecular spikes" all seem quite high e.g. > 50 counts (according to figure 1). Many in the field are interested in quantifying the transcript abundance of lowly expressed genes such as transcription factors. Can "molecular spikes" evaluate the accuracy of transcript quantification by different scRNA-seq protocols for such lowly expressed genes (ex. 1 to 10, 10 to 50 counts range)?

In Figure 1, the molecular spike-in was added at intentionally high abundance in order to cover even high expression levels, and subsampling was used to investigate molecule counting as a function of expression levels. Therefore, the initial analyses were informative over the full range of expression levels (e.g., Figure 2).

In the revised manuscript, we have extensively updated our spike-in design and created a new set of molecular spikes comprised of 11 distinct transcript sequences that span different lengths (500-3000 bp) and GC contents (40-60 %). Within each of these 11 different spike-in sequences, we included both the spUMI for ground-truth counting and also 24 specific 7 nt barcodes that encode for varying expression levels over 12 steps of 2-fold difference. Thus, within each distinct spike-in sequence we can create an expression standard curve. Elegantly, the expression levels of every barcode were accurately derived from counting using the spUMI, and were not relying on error-prone RNA measurements or cloning complexity estimates.

Reviewers Figure 1 (Figure 3a in the revised manuscript). (a) Illustration of the design of the complex set of molecular spike-ins. The set consists of 264 unique spike-ins based on 11 distinct spike sequences of different lengths and GC levels, as shown. Each of these 11 sequences further contains 24 different barcodes that were introduced in titration to obtain a standard curve per spike in sequence (using 2 barcodes per expression level, covering 12 different expression levels).

We further show that the resulting set of 264 spike-ins (24 barcodes x 11 sequences) span the observed expression levels in human cells well and model technical variability excellently (new Figure 3c, pasted below). These features and the fact that the exact molecular abundances of the spikes can be counted using the spUMIs makes them ideal as a replacement for the most commonly used ERCC spikes and for spike-in based normalization & biological variability classification.

Reviewers Figure 2 (Figure 3c in revised manuscript): (c) Scatter plot showing the mean molecules detected per cell (x-axis) against the squared CV (y-axis).

3. The authors note that "the reported increased sensitivity obtained in tSCR-seq is completely artificial due to the removal of the essential cDNA cleanup step." (Line 92). Is there a quantification to show that after correction with "molecular spikes", the sensitivity of tSCR-seq is comparable to the old SCR-seq and thus the previously reported increased sensitivity is "completely artificial"? Generally, given the strength of this statement, further quantitative substantiation is warranted.

In Figure 1 we showed that RNA counts reported with tSCR-seq is linearly following sequence depth, meaning that new UMIs are added in each new PCR cycle for each amplicon. Worse counting than that is hard to imagine.

Given that the tSCR-seq data we have presented in this manuscript was generated with the first version of molecular spikes, correction of the inflated counts to quasi-UMIs (as we demonstrate in Figure 3) is not possible. This is because realistic parameter estimates for the Poisson-lognormal distribution of quasi-UMIs require molecule count data over the 264 new spike-ins that span a wide expression range are needed. In line with the reviewer's comment, we have modified the statement on the tSCR-seq findings to more accurately reflect what can be from the data.

"Thus, the reported improvement in sensitivity in tSCR-seq¹¹ is completely artificial since the reportedly increased UMI counts do not correspond to RNA molecules."

4. If overcounting is identified using "molecular spikes" such as in the tSCR-seq benchmark, how do the authors suggest correcting for such overcounting? Does the correction need to be applied to the protocol stage or can be correction by achieved computationally? Does such overcounting always linearly follows sequencing depth such that the overcounted amount could be subtracted or regressed out?

Thank you for the insightful question, which inspired us to develop a correction strategy based on the molecular spike-ins. In the revised manuscript, we describe a computational correction strategy we developed and implemented for experiments where overcounting is indeed detected by molecular spikes. While in general corrections are indeed preferable at the protocol stage, any data generated using the new molecular spike-in set can also be used with our computational approach. To this end, we utilize a recently described quantile normalization strategy. Based on the molecule counts derived from the internal spUMI of the new molecular spike-in set, we generate accurate maximum-likelihood estimates of Poisson-lognormal distribution parameters. We then use the data-driven shape parameter to normalize inflated or read-count based data (e.g., in the case that lacks UMIs completely) to normalize such counts to quasi-UMIs that closely resemble real UMIs (new Figures 3h,i,j, also pasted below).

Reviewers Figure 3 (Figure 3h,i,j in revised manuscript): (h) Distributions of non-zero raw reads, inferred quasi-UMIs and observed Smart-seq3 UMIs across genes, shown for 20 representative HEK cells. (i) Boxplots showing the total number of reads, inferred quasi-UMIs and observed Smart-seq3 UMIs per cell (n = 151). (j) Scatter plot of observed Smart-seq3 UMIs against inferred quasi-UMIs across all genes (n = 17,054) for a representative cell.

Minor comments:

1. Does correction using "molecular spikes" enable the discovery of new cell-types or cell-states that are not apparent without correction in the tSCR-seq data?

Unfortunately, the tSCR-seq data would have needed to be generated with the revised molecular spikes set to enable computational correction to quasi-UMIs. However, in line with findings of Townes & Irizarry 2020, we fully expect quasi-UMIs to outperform inflated read-counts in biological discoveries as virtually all modern analysis approaches from normalization to clustering and differential expression analysis are tuned towards UMI data (for example SCTransform, which explicitly states incompatibility with non-UMI data).

2. Do "molecular spikes" provide a better correction of transcript quantification than previously proposed computational correction strategies?

Molecular spikes are unique in containing an internal counting strategy to experimentally validate RNA molecule counting. Since all previous spike-ins relied on correlating UMI-based counts to approximately measured relative abundances, the molecular spikes for the first time open up the possibility to obtain a realistic correction strategy for transcripts that are quantified without UMIs (e.g., Smart-seq2).

Decision Letter, first revision:

Subject: AIP Decision on Manuscript NMETH-BC45850B
Message: Our ref: NMETH-BC45850B

28th Jan 2022

Dear Rickard,

I hope you had a good start of the new year. And thank you for submitting your revised manuscript "Molecular spikes: a gold standard for single-cell RNA counting" (NMETH-BC45850B). It has now been seen by the original referees and their comments are below. The reviewers find that the paper has improved in revision, and therefore we'll be happy in principle to publish it in Nature Methods, pending minor revisions to satisfy the referees' final requests and to comply with our editorial and formatting guidelines.

We are now performing detailed checks on your paper and will send you a checklist detailing our editorial and formatting requirements in about 10 days. Please do not upload the final materials and make any revisions until you receive this additional information from us.

TRANSPARENT PEER REVIEW

Nature Methods offers a transparent peer review option for new original research manuscripts submitted from 17th February 2021. We encourage increased transparency in peer review by publishing the reviewer comments, author rebuttal letters and editorial decision letters if the authors agree. Such peer review material is made available as a supplementary peer review file. Please state in the cover letter 'I wish to participate in transparent peer review' if you want to opt in, or 'I do not wish to participate in transparent peer review' if you don't. Failure to state your preference will result in delays in accepting your manuscript for publication.

Thank you again for your interest in Nature Methods Please do not hesitate to contact me if you have any questions.

Best regards,
Lei

Lei Tang, Ph.D.
Senior Editor
Nature Methods

ORCID

Reviewer #1:

Dear Authors and Editor,

I would like to recommend the publication of the study "Molecular spikes: a gold standard for single-cell RNA counting" by Ziegenhain et. al. The manuscript has markedly improved with the addition of expanded spike-in designs that are reduced to practice with a single-cell experiment. Notable new contributions include:

1. Expanded design of molecular spike-in controls, including 264 molecular spikes (with inbuilt spUMIs) across different expression abundances.
2. Provision of a computational strategy and software package to facilitate spike-in analysis (R package UMImcountR).
3. Demonstrate the use and advantages of the spike-ins within a single-cell experiment (HEK293FT cells generated with Smartseq3xpress protocol).

I can foresee molecular spike-ins being widely useful within single-cell experiments, as well as other quantitative NGS assays. The spike-ins allow RNAs to be correctly counted in the absence of UMIs or when UMI-based counting is biased. This can be used to improve the technical performance of single-cell sequencing. Widespread adoption will enable easier comparisons between experiments, samples and different methods. I also think the ability to more accurately measure differences in mRNA amounts between different cell types, tissues and sizes could lead to some novel biological insights. I prefer the use of a more exact name to 'molecular spikes'; since spike-in controls are also used DNA and protein (also molecules, albeit not single-molecules) but appreciate this is the authors prerogative.

Congratulations on an improved manuscript, well-developed study, and an insightful advance with wide benefits.

Reviewer #3

I thank the authors for this substantial revision. The revised text has now better clarified the difference between molecular spikes versus RNA spike-in pools. The new Figure 3 and associated discussions are great additions in this revision and better demonstrates the effectiveness of molecular spikes in guarding against overcounting. The new accompanying R software package for correcting for overcounting using molecular spikes also makes it potentially more useful to the scientific community.

I have the following set of minor comments:

1. I would also encourage the authors to include the code used to design the complex set of molecular spike-ins available in case others are interested in using this process for designing molecular spikes for other organisms.
2. It remains somewhat unclear to me how the authors envision the use of molecular spikes will enable discoveries. Are there new cell-types or cell-states that we anticipate may not be visible due to overcounting? Alternatively, is the main utility of molecular spikes to serve as a control in the development of new approaches like tSCRB-seq? A discussion on the potential utility for precise RNA counting coupled with full transcript characterization by SmartSeq protocols for example would greatly clarify the utility of molecular spikes and encourage its adoption.

Author Rebuttal:

Reviewer #2:

Remarks to the Author:

The authors have substantially extended their work, enriching the spike in sequence and concentration characteristics. The revised manuscript shows that such spike ins can be helpful in diagnosing UMI problems, adjusting read counts without the use of UMIs, and estimating cell size variation.

We thank the reviewer for appreciating the new results added during the revision.

I am a bit puzzled by the result showing that cell-to-cell variation in spike in abundance provides a reasonable base-level for evaluating gene variability (Fig. 3c). I would have thought that variation in

lysis and other technical aspects prior to RT would place spike-in variation lower than the background gene variation. I can't help but wonder whether the specific experiment being analyzed in Fig. 3c would be representative of less perfect conditions (e.g. primary tissue sample, etc.). To be clear, I don't see any problem with the presented result. But it would be helpful for the readers if the authors commented on this issue.

We have amended the text with a qualifier on this matter:

“Contrasting the mean-variance relationship for endogenous genes and the 264 spike-ins confirmed that the set of molecular spikes spanned relevant endogenous expression levels and that they accurately modelled technical variation in the homogenous HEK cell population (**Figure 3c**).”

Minor concerns:

- Fig. 3e and related text. It's a bit confusing as the authors switch between number of reads and number of molecules (UMIs). What is the relationship between % of spike-in molecules and cell size? If the % of spike-in reads shows better correlation with size, what would be a plausible explanation for that?

We apologize for the confusion. During the data analysis we have checked both the % spike-in reads as well as the % spike-in molecules/UMIs in relation to the cell size. Both analyses showed equally high correlation in our dataset. We opted to present and suggest for users to analyze the %spike-in reads, since that analysis is independent of the library UMI counting strategy and accuracy.

- Also, the reference to “strongly correlated with sequence depth” on line 159 could be a bit misleading, as “sequence depth” commonly refers to the sequencing depth of the whole library. Whereas here, the authors simply refer to the total number of reads per cell, which is how it referred to in Supp. Fig. 7e-f, and something that the total UMI counts are naturally expected to correlate with.

Good point, we have adjusted the wording accordingly:

“We then investigated to what extent the total detected RNAs per cell followed the cell diameter, which showed only modest correlation (**Figure 3d**; Spearman $r=0.34$), whereas total RNAs detected per cell strongly correlated with the number of sequenced reads.”

- line 185, Fig. 3j – what was the correlation? Was it significantly higher than that of uncorrected counts? Please show statistics.

We have added the correlation coefficient (spearman's $\rho=0.93$) to the main text. The correlation coefficient was similar to that of uncorrected counts, which makes sense because the quasi-UMI method provides a rescaling but should keep relative abundances intact.

Final Decision Letter:

Subject: Decision on Nature Methods submission NMETH-A45850C
Message:

6th Mar 2022

Dear Professor Sandberg,

I am pleased to inform you that your Article, "Molecular spikes: a gold standard for single-cell RNA counting", has now been accepted for publication in Nature Methods. Your paper is tentatively scheduled for publication in our May print issue, and will be published online prior to that. The received and accepted dates will be 20th Apr 2021 and 6th Mar 2022. This note is intended to let you know what to expect from us over the next month or so, and to let you know where to address any further questions.

Your paper will now be copyedited to ensure that it conforms to Nature Methods style. Once proofs are generated, they will be sent to you electronically and you will be asked to send a corrected version within 24 hours. It is extremely important that you let us know now whether you will be difficult to contact over the next month. If this is the case, we ask that you send us the contact information (email) of someone who will be able to check the proofs and deal with any last-minute problems.

If, when you receive your proof, you cannot meet the deadline, please inform us at rjsproduction@springernature.com immediately.

Once your manuscript is typeset and you have completed the appropriate grant of rights, you will receive a link to your electronic proof via email with a request to make any corrections within 48 hours. If, when you receive your proof, you cannot meet this deadline, please inform us at rjsproduction@springernature.com immediately.

Once your paper has been scheduled for online publication, the Nature press office will be in touch to confirm the details.

Content is published online weekly on Mondays and Thursdays, and the embargo is set at 16:00 London time (GMT)/11:00 am US Eastern time (EST) on the day of publication. If you need to know the exact publication date or when the news embargo will be lifted, please contact our press office after you have submitted your proof corrections. Now is the time to inform your Public Relations or Press Office about your paper, as they might be interested in promoting its publication. This will allow them time to prepare an accurate and satisfactory press release. Include your manuscript tracking number NMETH-A45850C and the name of the journal, which they will need when they contact our office.

About one week before your paper is published online, we shall be distributing a press release to news organizations worldwide, which may include details of your work. We are happy for your institution or funding agency to prepare its own press release, but it must mention the embargo date and Nature Methods. Our Press Office will contact you closer to the time of publication, but if you or your Press Office have any inquiries in the meantime, please contact press@nature.com.

If you are active on Twitter, please e-mail me your and your coauthors' Twitter handles so that we may tag you when the paper is published.

Please note that Nature Methods is a Transformative Journal (TJ). Authors may publish their research with us through the traditional subscription access route or make their paper immediately open access through payment of an article-processing charge (APC). Authors will not be required to make a final decision about access to their article until it has been accepted. Find out more about Transformative Journals

Authors may need to take specific actions to achieve compliance with funder and institutional open access mandates. If your research is supported by a funder that requires immediate open access (e.g. according to Plan S principles) then you should select the gold OA route, and we will direct you to the compliant route where possible. For authors selecting the subscription publication route, the journal's standard licensing terms will need to be accepted, including self-archiving policies. Those licensing terms will supersede any other terms that the author or any third party may assert apply to any version of the manuscript.

To assist our authors in disseminating their research to the broader community, our SharedIt initiative provides you with a unique shareable link that will allow anyone (with or without a subscription) to read the published article. Recipients of the link with a subscription will also be able to download and print the PDF. As soon as your article is published, you will receive an automated email with your shareable link.

Please note that you and your coauthors may order reprints and single copies of the issue containing your article through Nature Research Group's reprint website, which is located at <http://www.nature.com/reprints/author-reprints.html>. If there are any questions about reprints please send an email to author-reprints@nature.com and someone will assist you.

Best regards,
Lei

Lei Tang, Ph.D.
(she/her/hers)
Senior Editor
Nature Methods